# Learning Neuro-Symbolic Skills for Bilevel Planning

**Tom Silver, Ashay Athalye**
**Joshua B. Tenenbaum, Tomás Lozano-Pérez, Leslie Pack Kaelbling**
MIT Computer Science and Artificial Intelligence Laboratory
{tslvr, ashay, jbt, tlp, lpk}@mit.edu

**Abstract:** Decision-making is challenging in robotics environments with continuous object-centric states, continuous actions, long horizons, and sparse feedback. Hierarchical approaches, such as task and motion planning (TAMP), address these challenges by decomposing decision-making into two or more levels of abstraction. In a setting where demonstrations and symbolic predicates are given, prior work has shown how to learn symbolic operators and neural samplers for TAMP with manually designed parameterized policies. Our main contribution is a method for learning parameterized polices in combination with operators and samplers. These components are packaged into modular *neuro-symbolic skills* and sequenced together with search-then-sample TAMP to solve new tasks. In experiments in four robotics domains, we show that our approach — bilevel planning with neuro-symbolic skills — can solve a wide range of tasks with varying initial states, goals, and objects, outperforming six baselines and ablations. Video: https://youtu.be/PbFZP8rPuGg Code: https://tinyurl.com/skill-learning

**Keywords:** Skill Learning, Neuro-Symbolic, Task and Motion Planning

## 1 Introduction

Decision-making in robotics environments with continuous state and action spaces is especially challenging when tasks have long horizons and goal-based objectives. A common strategy is to decompose decision-making into a high level ("what to do") and a low level ("how to do it") [1, 2, 3]. Seminal work by Konidaris et al. [4] considers such a hierarchy where symbolic AI planning [5] is used to sequence together temporally-extended continuous skills [6]. Their main contribution is a method for learning symbols from known skills. In this work, we study the inverse: learning skills from known symbols [7, 8, 9, 10, 11]. We are motivated by cases where it is easier to design [12, 13, 14, 15, 16] or learn [17, 18, 19, 20, 21] symbols than it is to design skills. In the long term, we envision a continually learning robot that uses symbols to learn skills and vice versa.

We consider symbols in the form of *predicates* — discrete relations between objects. A set of predicates induces a state abstraction [3, 22] of a continuous object-centric state space. For example, consider the *Stick Button* environment (Figure 1, third column), where a robot must press buttons either with its gripper, or by using a stick as a tool. Predicates include Grasped and Pressed, inducing an abstract state such as {Grasped(robot,stick), Pressed(button1), ...}. We leverage symbolic predicates to learn skills that generalize substantially from limited demonstration data. Concretely, we learn skills that achieve certain symbolic *effects* (e.g., Grasped(robot, stick)) from states where certain symbolic *preconditions* hold (e.g., HandEmpty(robot)).

Symbolic scaffolding for skill learning has two major benefits. First, by drawing on AI planning techniques, we can efficiently chain together long sequences of learned skills [4, 5, 10]. This benefit is key in our setting where planning in the low-level transition space is practically infeasible, even though a black-box transition function is deterministic and known — actions are continuous, solutions can exceed 100 actions, and there is no extrinsic feedback available before a task goal is reached. Second, because each skill acts in a confined region of the state space, learning is easier than it would be for a monolithic goal-conditioned policy (e.g., with behavioral cloning [23]).

6th Conference on Robot Learning (CoRL 2022), Auckland, New Zealand.

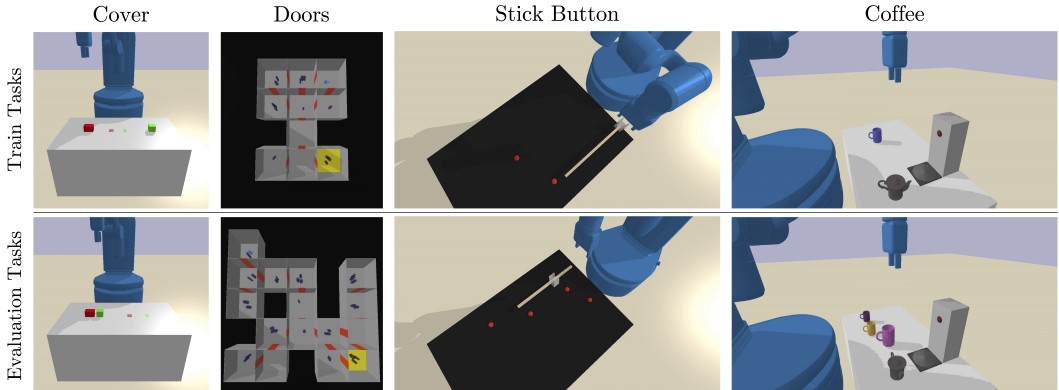

Figure 1: **Environments**. Top row: train task examples. Bottom row: evaluation task examples. See (§6) for descriptions of each environment.

While symbols offer useful structure for skill learning and planning, they also present challenges. In particular, symbolic state abstractions are often *lossy* [12, 13, 24], in that they discard information that may be important for decision-making, like the reachability of buttons or the relative stick grasp in Stick Button. This lossiness can be mitigated to some degree by inventing new predicates [4, 21], but in robotics environments, there are often kinematic and geometric constraints that are hard to perfectly abstract away [1, 25, 26]. These difficulties lead us to the following key desiderata.

**Key Desideratum 1 (KD1).** Because the abstraction is lossy, some environment states that correspond to the same symbolic state may be better than others in terms of completing a task. Thus, *a skill should be able to reach many different environment states ("subgoals") that correspond to the same abstract state.* For example, a skill that achieves Grasped(robot, stick) should not always lead to the same relative grasp: the grasp should be lower when buttons are farther away, and higher if collisions with the gray stick holder would prevent a bottom grasp. This desideratum implies that lookahead may be necessary, where the choice of skill early in the plan should be influenced by soft or hard constraints later in the plan. It also implies that we should not rely on the (abstract) subgoal condition in the framework of Konidaris et al. [4].

**Key Desideratum 2 (KD2).** When the state abstraction is lossy, it is not always possible to take actions in the environment that correspond to a particular abstract state sequence. Thus, *an agent should be able to consider multiple skill sequences that reach the same goal from the same initial abstract state.* For example, without a predicate that reflects button reachability, the agent may first plan to press all buttons without the stick, but if some buttons are unreachable, it will need the stick.

To address these two key desiderata, we propose an approach for learning and planning with *neuro-symbolic skills*. A neuro-symbolic skill consists of a symbolic operator [27], a neural subgoal-conditioned policy, and a neural subgoal sampler [28, 29]. The subgoals produced by the sampler and consumed by the policy correspond to different environment states that are consistent with the abstract effects of the operator, addressing KD1. The symbolic operators enable a bilevel planning algorithm [30, 31, 32], with AI planning in an outer loop and sampling in an inner loop. If sampling fails for an operator sequence, the outer loop continues to another sequence, addressing KD2. Skills are learned from demonstrations without any prior knowledge about the number of skills.

**Main Contributions.** Prior works by Silver et al. [12] and Chitnis et al. [13] propose a pipeline for learning operators and samplers for bilevel planning when *given* a set of parameterized policies[1]. While some policies may be human-designed and generally reusable, e.g., navigation policies that use motion planners, others are more domain-specific, such as pouring a pot of coffee or opening a door (Figure 1). Our main contribution is a method for learning these policies to complete the neuro-symbolic skills. We conduct experiments in four robotics environments (Figure 1) that require a wide range of behaviors, such as pouring, opening, pressing, picking, and placing. We find that skills learned from 100–250 demonstrations generalize to novel states, goals, and objects. Furthermore, our approach — bilevel planning with neuro-symbolic skills (BPNS) — outperforms six baselines and ablations, including learning graph neural network-based metacontrollers [15, 33].

---

[1]See (§C.3) for an elaborated discussion on the relationship between this work and [12, 13].

## 2 Problem Setting

We consider the problem of learning from demonstrations in deterministic, fully-observed environments with object-centric states, continuous actions, and a known transition function. An *environment* is characterized by a tuple $\langle \Lambda, \mathcal{U}, f, \Psi \rangle$. An object *type* $\lambda \in \Lambda$ has a name (e.g., `button`, `robot`) and a tuple of real-valued features (e.g., $(x, y, z, radius, color, \dots)$) of dimension $\dim(\lambda)$. An *object* $o \in \mathcal{O}$ has a name (e.g., `button1`) and a type, denoted $\mathsf{type}(o) \in \Lambda$. A *state* $x \in \mathcal{X}$ is an assignment of objects to feature vectors, that is, $x(o) \in \mathbb{R}^{\dim(\mathsf{type}(o))}$ for $o \in \mathcal{O}$. The *action space* is denoted $\mathcal{U} \subseteq \mathbb{R}^m$. The *transition function* is a known, deterministic mapping from a state and action to a next state, denoted $f : \mathcal{X} \times \mathcal{U} \to \mathcal{X}$. A *predicate* $\psi \in \Psi$ has a name (e.g., `Touching`) and a tuple of types (e.g., `(stick, button)`). A *ground atom* is a predicate and a mapping from its type tuple to objects (e.g., `Touching(stick, button1)`). A *lifted atom* instead has a mapping to typed variables, which are placeholders for objects (e.g., `Touching(?s, ?b)`). Predicates induce a state abstraction: $\mathsf{abstract}(x)$ denotes the set of ground atoms that hold true in $x$, with all others assumed false. We use $s \in \mathcal{S}$ to denote an abstract state, i.e., $\mathsf{abstract} : \mathcal{X} \to \mathcal{S}$.

An environment is associated with a *task distribution* $\mathcal{T}$, where each $T \in \mathcal{T}$ is characterized by a tuple $\langle \mathcal{O}, x_0, g, H \rangle$. The set of objects in the task is denoted $\mathcal{O}$. Importantly, objects vary between tasks. The initial state of the task is denoted $x_0 \in \mathcal{X}$. The goal, a set of ground atoms with predicates in $\Psi$ and objects in $\mathcal{O}$, is denoted $g$. A goal represents a set of abstract states: for example, there are many possible abstract states where $g = \{\texttt{Pressed(button2)}\}$ holds. The task horizon, the limit for how many actions can be taken in a task, is denoted $H \in \mathbb{Z}^+$. A *solution* to a task is a sequence of actions that achieves the goal and does not exceed the task horizon, i.e. $\overline{u} = (u_1, \dots, u_k)$ such that $k \le H$, $g \subseteq \mathsf{abstract}(x_k)$, and $x_i = f(x_{i-1}, u_i)$ for $1 \le i \le k$.

We consider a standard learning setting where *train tasks* drawn from from $\mathcal{T}$ are available at training time, and held-out *evaluation tasks* drawn from $\mathcal{T}$ are used for evaluation. Our objective is to maximize the number of evaluation tasks solved within a wall-clock timeout. Each train task $T$ is associated with one demonstration $\langle T, \overline{u}, \overline{x} \rangle$, where $\overline{u}$ is a solution for $T$ and $\overline{x} = (x_0, \dots, x_k)$ is the sequence of states visited. In experiments, tasks are randomly sampled, and demonstrations are generated with handcrafted environment-specific policies (but see (§C.4)).

## 3 Neuro-Symbolic Skills

How can we leverage the available demonstrations, predicates, and known transition model to maximize the number of evaluation tasks solved before timeout? We propose to learn and plan with neuro-symbolic skills. In this section, we define these skills formally; in (§4), we discuss planning; and in (§5), we address learning. See Figure 2 for a summary of the architecture.

A *skill* is a tuple $\phi = \langle \overline{v}, \omega, \pi, \sigma \rangle$ where $\overline{v}$ is a tuple of *arguments*; $\omega$ is an *operator*; $\pi$ is a *subgoal-conditioned policy*; and $\sigma$ is a *subgoal sampler*. A set of skills is denoted $\Phi$. Arguments are variables with types in $\Lambda$ and are shared by $\omega$, $\pi$, and $\sigma$. An *operator* is a tuple $\omega = \langle \overline{v}, P, E^+, E^- \rangle$ where $P, E^+, E^-$ are *preconditions*, *add effects*, and *delete effects* respectively, each a set of lifted atoms over the predicates $\Psi$ and arguments $\overline{v}$. A *subgoal-conditioned policy* is a mapping $\pi : \mathcal{O}^{|\overline{v}|} \times \mathcal{X} \times \mathcal{X} \to \mathcal{U}$ that takes as input a tuple of objects for the arguments $\overline{v}$, the current state $x$, and a subgoal state $x'$, and maps it to an action $u$. A *subgoal sampler* is a generator of elements from a distribution of subgoal states conditioned on object arguments and a current state, that is, $\sigma : \mathcal{O}^{|\overline{v}|} \times \mathcal{X} \to \Delta(\mathcal{X})$. A *ground skill* is a skill with objects that match its arguments, denoted by a tuple $\underline{\phi} = \langle \phi, \overline{o} \rangle$ where $\overline{o}$ is a tuple of objects in $\mathcal{O}$ with types matching $\overline{v}$. (We use underlines to denote grounded entities.) The corresponding *ground operator*, *ground policy*, and *ground sampler* are defined analogously,

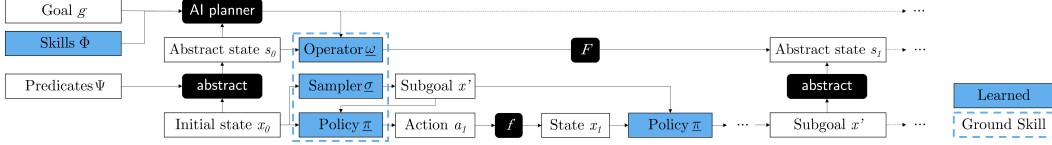

Figure 2: **Architecture**. Given learned symbolic operators, an AI planner is used to generate a candidate sequence of abstract states and skills. For each skill in the sequence, the learned sampler proposes a specific subgoal state in the next abstract state for the learned policy to pursue.

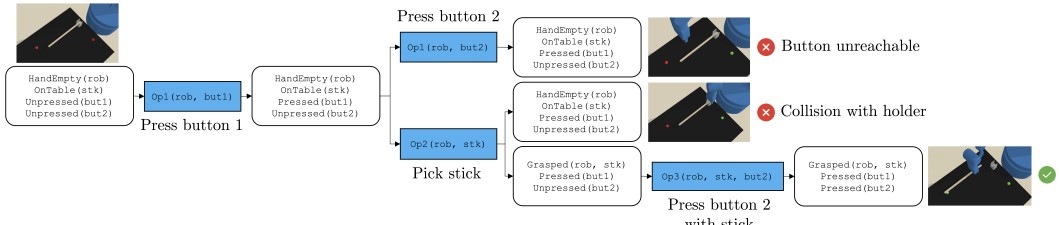

Figure 3: **Planning example**. First, an abstract plan that directly presses both buttons fails. Later, an abstract plan that uses the stick initially fails due to collisions, but succeeds on the next sample.

each associated with objects that match their typed variables, denoted $\underline{\omega} = \langle \underline{P}, \underline{E}^+, \underline{E}^- \rangle$, $\underline{\pi}$, and $\underline{\sigma}$ respectively. A set of ground skills $\underline{\Phi}$ induces an *abstract transition function* $F : \mathcal{S} \times \underline{\Phi} \to \mathcal{S}$ over the abstract state space through the add and delete effects of operators, given by $F(s, \underline{\phi}) = (s \setminus \underline{E}^-) \cup \underline{E}^+$, defined only when $\underline{P} \subseteq s$ (where $\setminus$ is set subtraction).

The contract between a skill's operator, policy, and sampler is the following: in states where the preconditions of a ground skill hold, i.e., $\underline{P} \subseteq \text{abstract}(x)$, the sampler should generate a subgoal $x' \sim \underline{\sigma}(x)$ that satisfies the skill's effects, i.e. $F(\text{abstract}(x), \underline{\phi}) = \text{abstract}(x')$, and the actions proposed by the policy $\underline{\pi}(\cdot, x')$ should lead to $x'$ from $x$. In practice, we *initiate* a ground skill only when the ground operator preconditions hold, and *terminate* at corresponding abstract successor state (cf. options [6]). Next, we show how to leverage this contract for efficient planning.

## 4 Bilevel Planning with Neuro-Symbolic Skills

Given skills and a new task, we search for a solution with bilevel planning. See Figure 3, Algorithm 1, and [13] for an extended description. Bilevel planning begins with an outer loop (Line 2), where skill operators are used to generate up to $N_{\text{abstract}}$ *abstract plans*. An abstract plan for task $T = \langle \mathcal{O}, x_0, g, H \rangle$ is a sequence of ground skills that achieves the goal, i.e. $\overline{\phi} = (\underline{\phi}_1, \ldots, \underline{\phi}_\ell)$ such that $s_0 = \text{abstract}(x_0)$, $g \subseteq s_\ell$, $\ell \leq H + 1$, and $s_i = F(s_{i-1}, \underline{\phi}_i)$ for $1 \leq i \leq \ell$.

```
plan(T, Φ, f)
    // Params:  N_abstract, H_skill, N_samples
1   s_0 ← abstract(x_0)
    // Uses operators
2   for φ̄ in top-k(s_0, g, Φ, N_abstract) do
        // Uses samplers and policies
3       ū ← refine(φ̄, T, Φ, f, H_skill, N_samples)
4       if ū is not null then
            // Planning succeeded
5           return ū
    // Planning failed
6   return null
```

**Algorithm 1:** Bilevel planning.

Much work in the AI planning literature is devoted to efficiently generating these abstract plans, with most state-of-the-art techniques building on heuristic search. In experiments, we use A* search with the LMCut heuristic [34]. The AI planner details are not important for this work; we have also experimented with other planners and heuristics including GBFS, K* [35], FF [36], and various configurations of the Fast Downward planner [5]. Unlike typical AI planning, we continue the search after the first plan is found, generating a potentially infinite sequence of candidate plans (cf. top-$k$ planning [35, 37, 38]).

Given a candidate abstract plan, we use the samplers and policies to perform a backtracking search over action sequences (Line 3). For each ground skill in the abstract plan, we use the ground sampler to produce a subgoal, and simulate the ground policy until either (1) the abstract transition is completed; or (2) a maximum skill horizon $H_{\text{skill}}$ is reached. In case (1), we continue on to the next ground skill, or terminate if the goal is reached (Line 5), while in case (2) we backtrack up to $N_{\text{samples}}$ times per step. If backtracking fails, we continue on to the next abstract plan [2].

---

[2]This planning strategy is not probabilistically complete, but see [32, 39] for complete variations.

# 5 Learning Neuro-Symbolic Skills from Demonstrations

We now address the problem of learning skills $\Phi$ from a dataset of demonstrations $\mathcal{D} = \{\langle T, \overline{u}, \overline{x} \rangle\}$. Recall that the data comprise raw sequences of states and actions for entire tasks. We preprocess the data in three steps: segmenting trajectories temporally, partitioning the segments into *skill datasets*, and lifting the objects within each skill dataset to variables. Then, for each skill dataset, we learn an operator, a subgoal-conditioned policy, and a subgoal sampler. We detail these steps below. This pipeline extends that of Chitnis et al. [13], who assume parameterized policies are given.

## 5.1 Data Preprocessing

**Segmentation.** For each demonstration $\langle T, \overline{u}, \overline{x} \rangle \in \mathcal{D}$, we identify a set of switch points $1 < i_1 < \cdots < i_\ell < |\overline{u}|$. Each pair of consecutive switch points $(j, k)$ induces a *segment* $\gamma = \langle \overline{u}_{j:k}, \overline{x}_{j-1:k} \rangle$, where $\overline{u}_{j:k} = (u_j, u_{j+1}, \ldots, u_{k-1})$ is a subsequence of actions from $\overline{u}$ and $\overline{x}_{j-1:k}$ is defined analogously. Temporal segmentation is a much studied problem in the literature [40, 41, 42]. In this work, inspired by the notion of contact-based modes (e.g., [43]), we create a new switch point whenever there is a change in the truth-value of any contact-related ground atom (§A). Intuitively, using more (fewer) predicates here instead would lead to finer (coarser) temporal abstractions.

**Partitioning.** Next, we partition the set of all segments into a collection of *skill datasets*. We group segments into the same skill dataset if their abstract effects are equivalent up to object substitution. Formally, given a segment $\gamma = \langle \overline{u}_{j:k}, \overline{x}_{j-1:k} \rangle$, let $s = \mathsf{abstract}(x_{j-1})$ and $s' = \mathsf{abstract}(x_{k-1})$. The *effects* of the segment are $\langle e^+, e^- \rangle = \langle s' \setminus s, s \setminus s' \rangle$. The *affected object set* $\mathcal{O}^e$ comprises all objects that appear in either $e^+$ or $e^-$. Two segments $\gamma_1, \gamma_2$ with effects $\langle e_1^+, e_1^- \rangle, \langle e_2^+, e_2^- \rangle$ and affected objects $\mathcal{O}_1^e, \mathcal{O}_2^e$ are *equivalent* ($\gamma_1 \equiv \gamma_2$) if there exists an injective mapping $\delta : \mathcal{O}_1^e \to \mathcal{O}_2^e$ such that $\delta(e_1^+) = e_2^+$ and $\delta(e_1^-) = e_2^-$ and where each object in $\mathcal{O}_1^e$ shares the type of the object it is mapped to in $\mathcal{O}_2^e$. The relation $\equiv$ induces the partition: each partition contains equivalent segments.

**Lifting.** The segments in each skill dataset involve different objects. When we perform lifting, we replace all the objects in a segment with variables that have the objects' types. For example, in Stick Button, the `robot` may use the `stick` to press `button1` in some segment, and the `robot` may use the `stick` to press `button2` in another segment. After lifting, these objects would be replaced with `?robot`, `?stick`, and `?button`. Formally, for each skill dataset $\Phi_i$, for an arbitrary segment $\gamma_0 \in \Phi_i$ with affected object set $\mathcal{O}_0^e$, we create one placeholder variable $v$ for each object in $\mathcal{O}_0^e$. We order these variables arbitrarily into a tuple $\overline{v}$. Using the injective mappings $\delta$ described above, it is then possible to map the affected object set for any segment in $\Phi_i$ to the variables $\overline{v}$. In addition to aligning segments with different objects, variables define the *skill scope*: when computing abstract transitions, subgoals, and actions, the skills will reference only the object states for those variables. The final output of preprocessing is a set of lifted skill datasets, i.e., skill datasets $\phi_i$, variables $\overline{v}$, and mappings from segment objects to variables that we will use in skill learning, described next.

## 5.2 Skill Learning

**Operator Learning.** Following previous work [13, 18, 21, 44], we use a simple linear-time approach to learn symbolic operators. We induce one operator from each lifted skill dataset $\Phi_i$. The variables $\overline{v}$ constructed via lifting comprise the operator arguments. The operator add and delete effects follow immediately from lifting a segment's effects $\langle e^+, e^- \rangle$: we replace all objects with the corresponding variables in $\overline{v}$, to arrive at lifted atom sets $\langle E^+, E^- \rangle$. To compute preconditions, we use all common lifted atoms in the initial segment states. For a segment $\gamma \in \Phi_i$ with states $\overline{x}_{j-1:k}$, let $s_\gamma = \mathsf{abstract}(x_{j-1})$, and let $s_\gamma^{\overline{v}}$ denote the corresponding lifted atom set, with all affected objects in $s_\gamma$ replaced with variables in $\overline{v}$, and with any ground atoms in $s_\gamma$ involving objects not in $\overline{v}$ discarded. The preconditions P are then the intersection of lifted atom sets for each segment, denoted $P = \bigcap_{\gamma \in \Phi_i} s_\gamma^{\overline{v}}$, completing the operator $\omega = \langle \overline{v}, P, E^+, E^- \rangle$.

**Policy Learning.** Recall the responsibility of a ground policy $\pi(x, x')$ is to output an action $u$ that will lead the agent closer to the subgoal $x'$ from the state $x$. At this point in the pipeline, for each skill, we have available a dataset of subgoals being achieved: the final state of each segment in the skill dataset is a subgoal, and the preceding states and actions demonstrate behavior towards that subgoal. We can therefore use these data for supervised learning of a subgoal-conditioned policy. First, we use the skill scope to convert the segment states into fixed-dimensional vectors by concate-

nating the features of the objects in the scope together with the subgoal. Formally, for each state $x$ in each segment $\gamma \in \Phi_i$, we construct a vector $x^{\overline{v}} = x(v_1) \circ \cdots \circ x(v_n)$, where $\overline{v} = (v_1, \ldots, v_n)$, $x(v_i) = x(o_i)$ for the corresponding $o_i$ under lifting, and $\circ$ denotes vector concatenation. We then create a dataset of input-output pairs $(x_{i-1}^{\overline{v}} \circ x_{k-1}^{\overline{v}}, u_i)$ for $j \leq i < k$ in each segment with states $\overline{x}_{j-1:k}$, where $x_{k-1}^{\overline{v}}$ is the segment subgoal. We are now left with the problem of learning a policy from vector-based supervised data. In this work, we use behavioral cloning with fully-connected neural networks (see (§B) for details), but many other choices are possible[3]. At evaluation time, if the policy is grounded with objects $\overline{o} = (o_1, \ldots, o_n)$, and queried in state $x$, then $x(o_1) \circ \cdots \circ x(o_n)$ is the state input to the neural network, while the subgoal input is generated by the ground sampler.

**Sampler Learning.** The role of the subgoal sampler is to propose different specific states, among the infinitely many consistent with the effects of the operator, for the policy to pursue. This flexibility is important for KD1 (§1). To learn samplers, we can largely reuse the datasets from policy learning: for each $(x \circ y, u)$ in the policy learning dataset, with $y$ denoting the subgoal, we create a $(x, y)$ input-output pair for sampler learning. Following [13], we learn two neural networks for each sampler. The first network outputs the mean and diagonal covariance of a Gaussian distribution. The second network is a binary classifier that takes a candidate $(x, y)$ sampled from the Gaussian as input and accepts or rejects. These two networks are capable of modeling a richer class of distributions than the Gaussian alone, e.g., multimodal distributions. See (§B) for architecture and training details[4].

We make two modifications to the approach described above to promote generalization. First, we use the *relative* subgoal $x_{k-1}^{\overline{v}} - x_i^{\overline{v}}$ for each step $i$ in the segment, rather than the absolute subgoal $x_{k-1}^{\overline{v}}$ (cf. goal relabelling [29, 46]). At evaluation time, we derive the absolute subgoal from the relative subgoal when the policy is first initialized, and update the relative subgoal after each transition. Second, we detect and remove subgoal dimensions that are static across all data (e.g., object mass).

**Limitations.** Our approach assumes that the given predicates comprise a useful state abstraction of the underlying state space with respect to the task distribution. With random or meaningless predicates, we would not expect to outperform non-symbolic baselines. However, previous work [12, 21] suggests that operator and sampler learning are robust to a limited degree to predicate ablation or addition; we expect the same for policy learning[5]. We assume that effective behavior can be determined from the skill scope alone and do not learn skills that handle a variable number of objects. Our method would create a separate skill for each unique skill dataset arising from affected object sets of different sizes, which might be preferred in some scenarios (picking up one, vs. ten, vs. two-hundred coins off the ground) but not others (tying multiple sticks together with a piece of string). Relational neural networks (e.g., GNNs) offer one possible path to address these two limitations. Through the lens of parameterized policy learning [47, 48, 49, 50], subgoals are just one form of parameterization among many possible alternatives. We also aspire to learn from *human* demonstrations on *real* robots; data efficiency results in the next section suggest that this is feasible.

## 6   Experimental Results

Our experiments address five questions about bilevel planning with neuro-symbolic skills (BPNS):

**Q1**. How many train tasks are required by BPNS to effectively solve held-out evaluation tasks?
**Q2**. Does BPNS generalize to unseen numbers of objects?
**Q3**. Can BPNS learn skills to complement existing general-purpose skills?
**Q4**. How important is the ability to sample multiple subgoals per abstract plan step (KD1)?
**Q5**. How important is the ability to generate multiple candidate abstract plans (KD2)?

**Environments.** We describe environments here at a high level, with details in (§A). The *Cover* environment was introduced by [12, 13, 21]. A robot must pick and place blocks to cover target regions on a 1D line. The robot can only initiate and release grasps in certain allowed regions. For example, if the robot grasps a block on the left side, it may not be able to place it to completely cover a target, depending on the allowed regions. Tasks have two blocks and two targets. There is random variation in the initial poses of the blocks, targets, allowed regions, and robot gripper.

---

[3]We also experimented briefly with implicit behavioral cloning [45] but did not notice any improvements.
[4]The policy and sampler for a skill are trained separately. In preliminary experiments, we observed that training them jointly sometimes performs better if training is stopped early, but is otherwise similar.
[5]See (§C.5) and (§C.6) for results with superfluous predicates and objects.

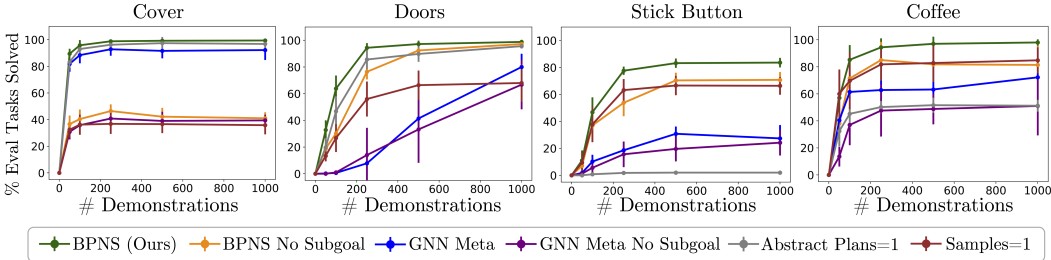

Figure 4: **Main results**. Evaluation task success rates versus number of demonstrations. Lines are means and error bars are standard deviation over 10 random seeds. GNN BC is excluded because its performance is consistently very poor, and its training is the most expensive; see the table below.

*Doors* is loosely inspired by the classic Four Rooms [6]. A robot must navigate to a target room while avoiding obstacles and opening doors. Doors have handles that require different rotations to open. In this environment, to address (**Q3**), a skill that moves the robot between configurations using a motion planner (BiRRT) is provided; door opening must be learned. Tasks have 4–25 rooms with random connectivity. The goals, obstacles, initial robot pose, and door parameters also vary.

The *Stick Button* environment is described in (§1). Train tasks have 1–2 buttons and evaluation tasks have 3–4 buttons. Tasks vary in the initial poses of the stick, holder, robot, and buttons.

In *Coffee*, a robot must move a pot of coffee to a hot plate, turn on the hot plate by pressing a button, and then move to pour the coffee into cups. The cups have different sizes and require different amounts of coffee. Depending on its orientation, the pot may need to be rotated before the handle can be grasped; the orientation is not reflected in the predicates. Overfilling a cup or pouring from too far results in a spill (failure). Train tasks have 1–2 cups and evaluation tasks have 2–3 cups. Tasks vary in the poses of the robot, pot, and cups, and in cup sizes. The goal is to fill all cups.

**Approaches.** We briefly describe the approaches we evaluate, with details in (§B).

- *BPNS*: Our main approach, bilevel planning with learned neuro-symbolic skills.
- *BPNS No Subgoal*: Our main approach, but with no samplers, and with subgoal-conditioned policies replaced with regular policies $\pi : \mathcal{X} \rightarrow \mathcal{U}$. This approach is inspired by [7, 9, 51].
- *GNN Meta*: Instead of AI planning, this approach learns a graph neural network (GNN) meta-controller [9, 15, 33] that outputs a next skill based on the current state, abstract state, and goal.
- *GNN Meta No Subgoal*: Same as GNN Meta, but without samplers, and with regular policies. This baseline is the closest to existing methods in the AI planning literature [9, 10, 51, 52].
- *GNN BC*: Learns a monolithic goal-conditioned GNN policy by behavioral cloning (BC).
- *Samples=1*: An ablation of BPNS where $N_{\text{samples}} = 1$ during planning.
- *Abstract Plans=1*: An ablation of BPNS where $N_{\text{abstract}} = 1$ during planning.

All approaches are trained with the same demonstration data and use the same predicates. All approaches also use the transition function $f$, except for GNN BC, which is model-free.

**Experimental Setup.** For all environments and approaches, we use 50 evaluation tasks and 10 random seeds. The timeout for each evaluation task is 300 seconds and the horizon $H = 1000$. Experiments were run on Ubuntu 18.04 using 4 CPU cores of an Intel Xeon Platinum 8260 processor.

**Results and Analysis.** Figure 4 shows performance as a function of demonstration count. BPNS performs well in all environments after 100–250 demonstrations (**Q1**). Stick Button is the most challenging because many considered abstract plans are infeasible, which can lead to timeouts. The strong performance in Stick Button and Coffee, which have more objects than seen during training, also allows us to affirmatively resolve (**Q2**). Strong performance in Doors, where motion planners are provided, confirms that BPNS is capable of incorporating general-purpose skills (**Q3**).

The importance of generating multiple samples per abstract plan step (**Q4**) is reflected in the No Subgoal baselines and the Samples=1 ablation. The ablation performs worse than BPNS in all environments, while the No Subgoal baselines perform worse in all except Doors with 1000 demonstrations. We found evidence for two explanations in our analysis. First, skill policies may fail to terminate with certain parameters, but succeed with others. Second, even if a policy terminates, the

agent may or may not be able to finish the rest of the task from that subgoal. These results confirm that KD1 (§1) is important for the strong performance of BPNS.

To analyze the importance of generating multiple candidate abstract plans (**Q5**), we can examine the GNN Meta baselines and the Abstract Plans=1 ablation. These approaches perform well in Cover and Doors, where the first abstract plan is usually refinable into a solution. In Coffee, the first abstract plan does not rotate the pot, and therefore fails when the handle is out of reach. See Figure 3 for an example of failure in Stick Button. These results confirm the importance of KD2 (§1).

We also compare BPNS with GNN BC, a purely model-free approach. The table on the right reports the mean (standard deviation) percent of evaluation tasks solved per environment after training with 1000 demonstrations. We

| *Environment* | **BPNS (Ours)** | **GNN BC** |
|---|---|---|
| Cover | 99.40 (0.64) | 4.60 (1.62) |
| Doors | 98.80 (0.80) | 9.20 (4.05) |
| Stick Button | 83.60 (1.78) | 0.20 (0.30) |
| Coffee | 98.00 (1.18) | 23.80 (4.76) |

also experimented with fewer demonstrations for GNN BC and saw consistently poor performance. The failure of GNN BC is unsurprising given the limited data and the high degree of task variability.

In the appendix, we present additional results that investigate the planning time of BPNS (§C.1), the poor performance of GNN Meta (§C.2), comparison to [12, 13] (§C.3), learning from human demonstrations (§C.4), the impact of irrelevant predicates (§C.5) or objects (§C.6), the filtering of low-data skills (§C.7), comparison to oracle skills (§C.8), the structure of the learned operators (§C.9), and skill learning with invented predicates (§C.10).

# 7 Related Work

Our work contributes to a long line of research in skill learning for robotics, with prior works considering both reinforcement learning [53, 54, 55, 56, 57, 58, 59, 60, 61, 62] and learning from demonstrations [8, 40, 63, 64, 65, 66, 67, 68, 33]. In the context of hierarchical RL [28, 29, 69], it is possible to interpret our skill samplers as managers and policies as workers. Our work is also related to *parameterized* skill learning [47, 70, 48, 71, 49, 50] if we view subgoals as parameters. In the AI planning literature, prior works use a set of given planning operators to learn skills [7, 9, 10, 72, 73, 51, 52, 74]. Achterhold et al. [75] pursue a similar approach in a robotics setting. Recent work by Guan et al. [15] is closest to our contribution in that they use given operators to learn skills with latent parameters that can be used to reach a diverse set of terminal states [61], and then learn a metacontoller over the skills. The "GNN Meta" baseline is inspired by this work. Instead of operators, recent works have also used pretrained language models to generate abstract "instructions" and then learn language-conditioned skills [68, 76, 77, 78].

Our work can also be viewed as learning components of a task and motion planning (TAMP) system. In this line, other work has concentrated on learning operators [79, 80, 81, 82, 44, 12, 83], learning samplers [39, 84, 85, 86, 13, 87], or learning predicates [17, 88, 4, 89, 90, 91, 18, 19, 20, 92, 21]. Our operator and sampler learning algorithms are most directly related to that of Chitnis et al. [13], who assume given parameterized policies. Driess et al. [26] use given operators to learn energy-based controllers from images in the context of constraint-based TAMP [43]. Wang et al. [86] learn to sample continuously parameterized skills to accomplish symbolic operator effects, such as pouring at a certain angle to fill a cup. To the best of our knowledge, our work is the first to learn operators, samplers, and policies for search-then-sample TAMP [30, 31, 32].

# 8 Conclusion

In this work, we proposed bilevel planning with learned neuro-symbolic skills as a framework for decision-making in object-centric robotics environments. We confirmed experimentally that this approach leads to strong generalization and data efficiency. There are many interesting open questions for future work. (1) Can neuro-symbolic skills be learned from invented predicates? In (§C.10), we offer preliminary evidence suggesting the viability of this direction. (2) Can neuro-symbolic skills be learned with reinforcement learning? (3) Can latent object feature learning [16] be used to drive skill learning? Answering these questions will help us leverage hierarchical planning at scale.

## Acknowledgements

We gratefully acknowledge support from NSF grant 1723381; from AFOSR grant FA9550-17-1-0165; from ONR grant N00014-18-1-2847; and from MIT-IBM Watson Lab. Tom is supported by an NSF Graduate Research Fellowship. Any opinions, findings, and conclusions or recommendations expressed in this material are those of the authors and do not necessarily reflect the views of our sponsors. We thank Rohan Chitnis, Nishanth Kumar, Willie B. McClinton, Aidan Curtis, Will Shen, and Anish Athalye for helpful comments on an earlier draft.

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
