# OpenReview forum: "Learning Neuro-Symbolic Skills for Bilevel Planning"
_robot-learning.org/CoRL/2022/Conference — CoRL 2022 Poster_

### Official Review · Reviewer_P5b1 · 2022-07-25

**Originality:** Very Good
**Technical Quality:** Very Good
**Clarity Of Presentation:** Excellent
**Impact:** 4

**Recommendation:**

Strong Accept: I recommend accepting the paper and will argue for my recommendation even if other reviewers hold a different opinion.

**Summary:**

This work proposes a method for learning sets of pddl-style operators along with their associated low-level control policies from demonstrations. These "neuro-symbolic skills" can then be used in a straightforward bi-level search to solve challenging task and motion planning problems. The approach assumes knowledge of a set of predicates which are sufficient to define an abstraction of the low-level state, as well as a deterministic dynamics model for the environment. However, where traditional TAMP approaches also require hand-engineered operators, and associated control policies, this work shows how these can be learned directly from the given demonstrations. Experiments show that this approach can be data efficient, and produce skills which can generalize to new test problems, containing unseen numbers/placements of objects.


**Issues:**

* I would like to see the performance of the learned operators compared to the hand engineered ones. I am curious whether the learned samplers would be more sample-efficient, given that they are trained on successful demonstrations.

**Quality Of The Limitations Section:**

Limitations are addressed clearly

**Reviewer Expertise:**

4: The reviewer is confident but not absolutely certain that the evaluation is correct

**Robotics Focus:**

Highly relevant to robotics but no hardware experiments

**Strengths And Weaknesses:**

Strengths:
* The idea of learning operators which can be used with AI planning techniques *and* associated low-level control policies is promising. The approach can plan very long sequences of actions to achieve temporally extended goals.
* I appreciate the similarity of the approach to existing sampling-based TAMP methods.
* The domain-agnostic treatment facilitated by PDDL and object-centric state makes it easy to see how this method could be applied to new domains and tasks.
* In general, I appreciate the clarity with which the approach is presented. And in particular, the thorough analysis of experimental results. The many ablations that the authors included helped shed light on the impact of the different aspects of the approach, as well as the difficulties inherent in each evaluation domain.

Weaknesses:
* I think that the idea that certain skills might be easier to describe in terms of symbols (as opposed to a low-level policy/controller) is interesting. However, I felt that the experiments didn't showcase this, since every demonstration is generated by a hand-engineered policy. It would really help sell the point if there was an experiment where the skill was more difficult to engineer, and demonstrations were provided by humans.
* A related point is that all the of the training data are generated by systems that themselves use the same set of predicates and bi-level search. This seems a strong prior, and makes for very straightforward segmentation of training demonstrations. And the segmentation appears to be crucial for the success of the overall approach. It would be more convincing if the demos came from a human or some other system that has no knowledge of those predicates.
* The assumption of fully-observable state is very strong, but I understand the problem is already hard enough, and hope to see this method extended to the more challenging and realistic partial observability.
* I think the choice of top-k planning seems like a weakness since the high level search for action skeletons isn't influenced by failures in the low-level policy. These failures should ideally help prune large portions of the high level search space.


**Summary Of Recommendation:**

I think the idea of using symbols as a scaffold for skill learning is interesting, and I like the fact that the learned operators allow these skills to be used by a standard PDDL planner to solve long-horizon tasks. The methodology described in this work is simple and clear, and the results are promising. Ideally, I would like to see this method used to learn from human demonstrations, but this is a good first step.

---

> ### Author Response · Authors · 2022-08-18
> **Thanks for your review!**
>
> Thanks very much for the encouraging review!
>
> >It would really help sell the point if there was an experiment where the skill was more difficult to engineer, and demonstrations were provided by humans… A related point is that all the of the training data are generated by systems that themselves use the same set of predicates and bi-level search… It would be more convincing if the demos came from a human or some other system that has no knowledge of those predicates.
>
> Thanks for this feedback. We conducted an additional experiment where demonstrations were provided by humans. Please see the additional experiments PDF in the meta-reviewer response.
>
> >The assumption of fully-observable state is very strong, but I understand the problem is already hard enough, and hope to see this method extended to the more challenging and realistic partial observability.
>
> Yes, we definitely agree. On the planning side, there has been some recent work on TAMP with partial observability (e.g., [1]). Extending our learning algorithms to handle partial observability (especially in terms of open worlds with unknown numbers of objects) is a very exciting direction for future work.
>
> >I think the choice of top-k planning seems like a weakness since the high level search for action skeletons isn't influenced by failures in the low-level policy. These failures should ideally help prune large portions of the high level search space.
>
> We very much agree, and actually mentioned this limitation in an earlier draft before removing it for space. To overcome this limitation, we could draw on techniques from the task and motion planning literature [2]. For example, [3] use discovered refinement failures to modify the operators and bias abstract planning away from repeating the same mistakes. Figuring out when and how the operators should be modified requires domain-specific rules in that work, but we are very interested in seeing whether similar rules can be learned automatically.
>
> >I would like to see the performance of the learned operators compared to the hand engineered ones. I am curious whether the learned samplers would be more sample-efficient, given that they are trained on successful demonstrations.
>
> We recorded these metrics and reported them in the additional experiment PDF attached to the meta-reviewer comment.
>
> [1] Garrett, C. R., Paxton, C., Lozano-Pérez, T., Kaelbling, L. P., & Fox, D. (2020). Online replanning in belief space for partially observable task and motion problems. ICRA.
>
> [2] Garrett, C. R., Chitnis, R., Holladay, R., Kim, B., Silver, T., Kaelbling, L. P., & Lozano-Pérez, T. (2021). Integrated task and motion planning. Annual review of control, robotics, and autonomous systems, 4, 265-293.
>
> [3] Srivastava, S., Fang, E., Riano, L., Chitnis, R., Russell, S., & Abbeel, P. (2014). Combined task and motion planning through an extensible planner-independent interface layer. ICRA.

---

> > ### Comment · Reviewer_P5b1 · 2022-08-27
> > **Reviewer response**
> >
> > Thank you for the clarifications. I appreciated the additional experiments, in particular the results of Table 4. However, I would suggest clarifying the meaning of the "Nodes Created" metric. It's not clear whether this refers to abstract search nodes, or sampling nodes created by the backtracking in the inner loop. Ideally both should be reported.

---

### Official Review · Reviewer_MQ6V · 2022-07-31

**Originality:** Good
**Technical Quality:** Very Good
**Clarity Of Presentation:** Good
**Impact:** 3

**Recommendation:**

Weak Accept: I recommend accepting the paper, but will not argue for my recommendation if the majority of other reviewers have a different opinion.

**Summary:**

This paper is about learning skills in task and motion planning problems via demonstrations. Learning a skill involves learning a goal-conditional policy together with a sub-goal sampler, as well as an operator that defines the logical state transition (postconditions and effects) that the skill will induce, assuming some preconditions are true when the skill is invoked by the task and motion planner.

The main contribution of the paper is showing how to learn policies, samplers, and operators from demonstration data that includes: the task definition (including a set of objects relevant for that task), the continuous state sequence, and the continuous controls. The continuous state transitions are segmented at contact points and thus split into skill dataset segments. These segments are lifted so that they are not grounded to particular objects, but types of objects. This is useful for learning general logical operators.

**Issues:**

The issues that I would love to see fixed in the revision would be W1 and W2. They would both make the paper's analysis and conclusions more robust, and more convincing in terms of the reliability of the skill learning method.

**Quality Of The Limitations Section:**

Limitations are addressed clearly

**Reviewer Expertise:**

4: The reviewer is confident but not absolutely certain that the evaluation is correct

**Robotics Focus:**

Highly relevant to robotics but no hardware experiments

**Strengths And Weaknesses:**

The paper is comprehensive and generally well-written. Like many other related papers in this area, this paper loses some readability because of the heavy use of notation and definitions that are used to explain symbolic planning. This is unfortunately very hard to avoid, but I think the reader would benefit from a reduction of formal notation in the writing, if it is done in a way that increases intuition about the definitions. The use of a running example could help with this. Aside from writing, I would like to mention the following strengths (S) and weaknesses (W):

S1: The generalization results in the low-training data regime (a few hundred demonstrated trajectories) are interesting, especially as the number of object varies at test time.

S2: Long-horizon planning with learned skills seems like it works very well for TAMP, and this could definitely inform future design of such planners.

S3: Segmenting demonstration trajectories at contacts makes a lot of sense for pick-and-place tasks.

W1: Lifting of demonstration trajectories that act on specific objects to more general types of objects seems error-prone, and I would have loved to see a more thorough analysis about how many logical state transition errors are introduced in the demonstration data via this pre-processing step. It seems brittle.

W2: The tasks selected for evaluation seem a bit simple, with only a few objects at training time (1-3) for many environments and (1-2) more at test time. I would loved to have seen more complex sequences of sub-goals involving more objects in the training environments.



**Summary Of Recommendation:**

My recommendation for a weak accept is based on the strengths of the paper and particularly S1, namely generalization of learned skills, including goal-conditioned policies and logical transition operators, with a few hundred demonstrations. It's an interesting result.

---

> ### Author Response · Authors · 2022-08-18
> **Thanks for your review!**
>
> Thanks very much for the helpful review!
>
> >The paper is comprehensive and generally well-written. Like many other related papers in this area, this paper loses some readability because of the heavy use of notation and definitions that are used to explain symbolic planning. This is unfortunately very hard to avoid, but I think the reader would benefit from a reduction of formal notation in the writing, if it is done in a way that increases intuition about the definitions. The use of a running example could help with this.
>
> Thank you! We appreciate this feedback. Please see our general response to the meta-reviewer for steps that we will take to improve readability.
>
> >W1: Lifting of demonstration trajectories that act on specific objects to more general types of objects seems error-prone, and I would have loved to see a more thorough analysis about how many logical state transition errors are introduced in the demonstration data via this pre-processing step. It seems brittle.
>
> We want to clarify a few points here and make sure we understand your critique. First, the lifting of demonstration trajectories described in Section 5.1 does not imply that all objects are interchangeable. If two objects have different low-level features, then one could define predicates to distinguish them. For example, suppose we have a single-dimensional color feature where 0.5 is red and 1.0 is blue. If two objects are different only in that color, we could distinguish them with a predicate that has classifies to True when “color” is greater than 0.75. Given this predicate, after lifting, the two objects would still be treated as two distinct entities because they would differ in that predicate evaluation. Without that predicate, the two objects would be treated as equivalent. In general, the predicates determine object equivalence classes, i.e., they induce an abstraction of the low-level state. When predicates are defined judiciously, this abstraction can be extremely helpful for planning efficiency and generalization. When they are defined poorly, the abstraction could discard too much information, or make distinctions between objects that would ideally be treated as equivalent. See L222-226 for further discussion of this limitation, and see the experiments on irrelevant predicates for additional insights.
>
> >W2: The tasks selected for evaluation seem a bit simple, with only a few objects at training time (1-3) for many environments and (1-2) more at test time. I would loved to have seen more complex sequences of sub-goals involving more objects in the training environments.
>
> We are definitely interested in applying our approach in more complex environments and are actively pursuing this direction. However, we want to emphasize a few ways in which the environments in this work are already quite sophisticated and diverse:
>
> * Our simplest environment, Cover, is adapted from previous work [1, 2], but our version is substantially more challenging. The action space in the previous work was already defined in terms of high level “pick” and “place” controllers. Here, we instead have a low-level action space [dx, dy, dgripper]. Each “pick” or “place” now requires a sequence of several low-level actions, which leads to a much longer task horizon.
> * Our second environment, Doors, is inspired by the Four Rooms environment that is pervasive in the option discovery literature [3]. The classic environment is typically a discrete, fixed grid world. Our version is significantly more involved in several ways. First, we have continuous state and action spaces. Second, the fine-grained action space leads to a much longer task horizon. Third, our room arrangements are not fixed, but instead randomly generated with each new task. Fourth, we added doors in between the rooms and required the robot to open the doors.
> * Our third environment, Stick Button, requires a simple form of tool use. Even when all representations are manually designed, planning with tools remains very challenging. For example, we were inspired by [4], which does not use any machine learning. This environment also features considerable task variation in the poses of the buttons, stick, and stick holder, which in turn leads to variation in the abstract plans that must be used to solve the tasks.
> * Our fourth environment, Coffee, is the most complex in terms of the number of distinct operations that must be performed to accomplish each task: rotating the jug, picking the jug handle, placing it on the hot plate, pressing a button to turn on the hot place, picking the jug handle again, and repeatedly moving and pouring into each coffee cup. There is also considerable task variation in the initial poses of the jug, machine, and coffee cup, as well as in the size of the coffee cup.
>
> See also the supplemental video for visualizations. See also our experiments PDF attached to the meta-reviewer response for additional experiments that involve up to 100 extra objects.

---

> > ### Author Response · Authors · 2022-08-18
> > **References**
> >
> > [1] Silver, T., Chitnis, R., Tenenbaum, J., Kaelbling, L. P., & Lozano-Pérez, T. (2021). Learning symbolic operators for task and motion planning. IROS.
> >
> > [2] Chitnis, R., Silver, T., Tenenbaum, J. B., Lozano-Perez, T., & Kaelbling, L. P. (2022).
> >
> > [3] Sutton, R. S., Precup, D., & Singh, S. (1999). Between MDPs and semi-MDPs: A framework for temporal abstraction in reinforcement learning. Artificial intelligence.
> >
> > [4] Toussaint, M. A., Allen, K. R., Smith, K. A., & Tenenbaum, J. B. (2018). Differentiable physics and stable modes for tool-use and manipulation planning. R:SS.

---

> > > ### Comment · Reviewer_MQ6V · 2022-08-26
> > > **Reviewer response**
> > >
> > > Thank you for the clarifications and for addressing my concerns. I will be recommending that this paper be accepted.

---

### Official Review · Reviewer_Y3Ww · 2022-08-01

**Originality:** Fair
**Technical Quality:** Fair
**Clarity Of Presentation:** Poor
**Impact:** 3

**Recommendation:**

Weak Accept: I recommend accepting the paper, but will not argue for my recommendation if the majority of other reviewers have a different opinion.

**Summary:**

The proposed method tackles the TAMP abstraction problem in learning from demonstration by starting from a manually defined set of symbolic predicates and learning a parameterized policy in combination with symbolic operators and low level subgoal (operator parameter) samplers. There are simulated results.

"We learn skills that achieve certain symbolic effects (e.g., Grasped(robot, stick))
from states where certain symbolic preconditions hold (e.g., HandEmpty(robot))."

Human programmers provide
- a definition of what a low level state is
- a definition of what low level actions are
- the low level dynamics function
- demonstrations in terms of low level states and actions
- an ontology: values, features, and objects have types [This may be what breaks in the real world, but works in games.]
- predicates which retrieve the values of features. The values of all possible predicates are known. In some cases the predicates seem carefully chosen to make the task work.

**Issues:**

Things that can be fixed in a reasonable amount of time:

WRITING
- improve and clarify writing (see my comments on that)
- make the paper stand alone more effectively, by moving some definitions and examples from the supplementary materials to the paper.

IN SIMULATION
- add large numbers of irrelevant predicates and operators
- see what happens if you don't filter out rarely used operators
- see what happens if you don't reject out of bounds subgoals
- wider variety of simulated tasks - ideally non-symbolic tasks and not effectively pick and place: paint something? repair something with glue, tape, or string? shape something such as clay or dough?

EFFICIENT PERFORMANCE
- explore several abstract plans in parallel rather than serially. This would deal with the issue of wasting a lot of time exhaustively sampling bad plans, and still allow adequate sampling on other plans.

EXTEND APPROACH
- start with demonstrations alone, measured with low level sensors (not manually defined high level
perceptions or predicates) as suggested in C.4.

Bigger changes I don't expect to be made, but could be made
- real robot implementation

MORE COMMENTS ON WRITING (to get around stupid character limits in the web interface):

Overloading lambda for object type and real valued features is confusing

In partitioning grouping segments if their affects are the same “up to object
substitution” is unclear. Does this mean any object of any type can be substituted into a
predicate
or does it mean only different object instances within the same type can be
substituted.


**Quality Of The Limitations Section:**

Limitations are not well addressed

**Reviewer Expertise:**

4: The reviewer is confident but not absolutely certain that the evaluation is correct

**Robotics Focus:**

Highly relevant to robotics but no hardware experiments

**Strengths And Weaknesses:**

This paper is on arxiv, the identity of the authors is public information, and this review was done knowing who the authors are. https://arxiv.org/abs/2206.10680

STRENGTHS

I agree with the general philosophy of this paper
that reasoning, planning, and learning at the symbolic level as
well as continuous levels is useful.

Good introduction.

KD2: I agree considering alternative "strategies" or abstraction sequences
is an important research area.

CONCERNS (some philosophical)

I disagree with the philosophy
that abstractions should be correct representations of the underlying problem
and should be taken seriously. I prefer approaches where abstract plans are often not executable but
suggest where to search in lower level planning spaces.
In current robot dynamic legged locomotion, hierarchical control is used
and more abstract representations are typically simplified (kinematic, dynamic, actuator, ...)
models that approximate
the lower levels but the output of the simplified models cannot be exactly obeyed by the lower
levels. The simplified models "suggest" where the lower level planners should search.

I would argue that the failure of early symbolic AI approaches to
robotics had to do with
needing the abstract plan to be correctly executable.
The proposed sampling-based approach only
samples low-level plans that conform to the abstract plan. Low level
planning approaches that explore a region of the low level planning space
suggested but not limited by a high level plan may be more useful.
Similarly, using imperfect or loose analogies may be a more effective way
of making use of abstractions than treating an abstraction as a "correct"
representation of the original problem. Abstractions where abstract plans work in the abstract
but don't always work or are not executable in the real world, and vice versa, may be the norm.
The role of abstraction may be to merely suggest good
starting points or search directions for low level search.

Another issue with this paper is the manual feature selection.
As Patrick Winston often said, if you have the right features, the
problem is easy, and practically any method will work. If you don't
have the right features, then no method really works well.
I am concerned that this work is doomed to succeed, with small numbers
of carefully chosen relevant features. For example, CupFilled is created
and MADE A CONTACT EVENT to get the approach to work on pouring.
This trivializes that task.
Our experience with good old fashioned AI (GOFAI) suggests that
it has been difficult to find the implementable predicates (symbols) for
real world robotics in the past. Has anything changed?

Could the paper discuss when useful abstractions are likely to exist and be findable, before a problem is solved? Note that after a problem is solved it is often easier to find useful abstractions to solve it.
Is it true that a successful abstract plan (as in line 132) always exists
when a successful low level policy exists? This is not clear to me.
A related question is whether a search in abstract space is guaranteed
to find a successful low level policy that does exist.
What properties of abstractions must hold for this to be true?
It may be that the only abstraction that works is no abstraction at all,
S==X. Consider a parity function, for example.

I am concerned the evaluation tasks as implemented in the simulator
are toy problems.
Gripping objects and sticks, door/door-handle manipulation, using the stick as a tool, and pouring are all trivialized. Small numbers of relevant predicates are available for each task. Do sticks bend or break? Do stick tips slip? Does liquid splash during pouring? These are not
simple skills that we can provide low level policies for that work well in general. As mentioned before,
CupFilled is made a contact event to make pouring work. A simple change to the work is to make all predicates from all task available to each task.
Lots of irrelevant predicates should cause segmentation and partitioning to fail.

The same argument goes for object types and objects. A relatively small number of relevant object types and objects are defined for each task. Are furniture objects? Lights and light switches? Interior decorations like pictures and rugs? How about the floor, wall, and ceiling? How about dust bunnies?
How are the objects involved in a task known? Are they all the objects that were present in the corresponding demonstration? Or just the ones changed (which ignores useful objects like tables)?

I note that demonstrations are generated with handcrafted
environment-specific policies. If doing the demonstrations is a solved
problem, why are we bothering to learn to do these tasks?

All the tests/experiments are done in simulation. What issues
exist concerning the sim-to-real gap when applying these plans and skills on a
real robot?
Will this approach work in the real world? Will it
run out of gas, as previous symbolic approaches to robotics have.
Is this work doomed to succeed because of the tasks and environments chosen to work with and the use of "clean" simulations? Simulated and "symbolic" (often man-made) environments and systems
(such as the ones used as tasks in this work) easily lend themselves to symbolic representations. Messy real world and "natural" systems don't.

I note that object types seem to be pre-defined rather than learned.
It is known what a button is, and that it can be pressed. This is a second manual intervention, in addition to defining predicates, whose implications are not discussed in the paper.
The robot
cannot construct a new object and discover it is a button.
Analogies must be exact, rather than loose or suggestive.
This makes lifting easy and well defined. How is the case handled where an
object has properties like a button, but is not completely like
a button in all ways, or fit the type definition?

"If sampling fails for an operator sequence, the outer loop continues to another sequence, addressing KD2."
Much more information needs to be provided about when the system gives up on
trying to attain a particular subgoal. KD2 should be modified to say "an agent
should be able to EFFICIENTLY consider multiple skill sequences ...".
Considering alternative "strategies" or abstraction sequences is not all that useful if it is not efficient.
The planning algorithm proposed in section 4 is likely to be very inefficient.

A concern is the notion that there are discrete objects. The notion of object
as presented in the paper breaks down with liquids, granular materials,
and deformable materials like peanut butter and clay (plastic materials?).
It also does not handle materials that are consumed, created, or transformed during use
(food, fuel, cooking, ...) well. It is difficult to reason about object construction and
structural failure (fracture, breakage, wear, fatigue, ...).
What does it mean for an object to be wet? Glued or nailed to another
object?

This paper reminds me of early work in machine learning on
symbolic concept learning, such as https://axon.cs.byu.edu/~martinez/classes/778/Papers/Mitchell.ch2.pdf
That work was useful, but eventually superseded with statistical or
numerical methods. Will the same happen to this work?

I agree that segmenting trajectories at contact state changes makes sense.
However, it probably makes sense to have a hierarchy of segments.
At the lowest level one has segments A, B, C ...
The next level might have AB, BC, ...
The next level might have ABC, ...
This helps deal with the case that multiple contact changes lead to a
known state (ball rolls down an inclined plane into a V shaped corner or
onto a floor).
It always ends up at the bottom of the V or on the floor,
but the contact sequences may vary.

The learned subgoal sampler as designed is unlikely to explore out-of-distribution parameters. For example let's say it needs to rotate a bolt or screw in a screw, and in the evaluation/real-world scenario, it happens that it needs to exert way more force than it is used to, simply because there might be more friction or corrosion, how likely is it to apply out-of-ordinary/distribution force? It cannot reason about the possible reasons for the screw being stuck.

How is the "true" goal of a demonstration known? How are irrelevant actions ignored? For example, human breathing is usually ignored in demonstrations, but this is not obvious without strong priors.
How are changes in the environment not caused by the demonstrator ignored (water evaporates or leaks,
objects may move on their own or due to gravity, ...)?
Lots of irrelevant features (predicates) should break this ability to tell what is relevant in a demonstration.

Add and delete operators were designed for features that had true or false values, and does not seem like the most useful semantics for robots.
I don't see how add and delete operators handle cases where a value is incremented in the paragraph on operator learning starting on line 184.
For a feature x with a value 3.2, to add 1.9 to x one doesn't delete x and then add x with the new value.
We don't want x in E- and in E+.
It makes more sense to have a set M of features whose values are modified.
What is the semantics of <e+,e-> = <s'\s,s\s'> when values of features are modified?

SUGGESTIONS

It is much easier to read this paper AFTER reading the supplementary material.
It would be nice if that were not the case. Many comments below (written as I
read the paper first) ask for definitions and examples, which are provided in the
supplementary material. The paper does not adequately stand alone.
The examples of symbolic
plans and how the underlying primitives work in the supplement are really helpful. It would be great if one of them could be in the paper itself.

I don't understand why the term "neural" is used in this paper.
There is no connection to biological neurons. The neural sub-goal conditioned
policy can be represented by any parameterized function framework,
as line 205 points out.
A subgoal sampler is just something that proposes instances from
a density function, which again can be represented in many ways, either
with a parametric function approximator or a nonparametric representation
which retains all the instances in the training set and picks one or
locally interpolates to propose a candidate.
Is the word neural used because that is the current fad in function
approximation? Is there any evidence that the neural network architectures
used in this paper are superior to other function approximators for the
purposes they are used in this paper? There are only a relatively small
number of training examples (less than 10,000), so the argument for
deep neural networks does not really apply.
I would call what is proposed a hybrid-symbolic-and-numeric algorithm, or a hierarchical algorithm.

Sections 3, 4, and 5 would be easier to read if a simple example was used
throughout. Maybe navigation through a series of rooms, where each room
was an abstract state, and skills existed to find doors, open them, and
move to another room.

It might be useful to note that KD1 implies some lookahead so later (soft and hard) constraints are
taken into account. This is what trajectory optimization and backpropagation
in time do.

I can't figure out what the tasks are just from Figure 1.

An environment (vs. a task) has object types (button), but not specific objects
(button1, button2, ...)?
Or did you want to include the object space O in the environment tuple
< Alpha, O, U, f, Phi >?
The set of objects in a task should be a subset of the set of objects
in the environment (irrelevant objects should be able to be present).

In top-k planning, how are the candidate plans made non-trivially
different? Such a planner could generate plans that have trivial
differences.

The notation in line 165 may not be familiar to the reader.
To make your paper more accessible you should explain what <s' \ s,s \ s'>
means in this context.

165: Can a segment be mapped to more than one skill dataset?

Humans learn with 1 or a few demonstrations.
This approach seems to need hundreds of demonstrations. Why?

The paper is good at presenting technical definitions and details. It would be even better if intuition was given, as well as definitions in words.

Words like operator, ground atom, and lifted atom seem to be necessary to understanding sections 3-5, but aren't explained until section 5.

How is this different from various other hierarchical approaches in
machine learning that simultaneously learn parameterized skills and/or
how to sequence them? Perhaps an explanation of how the components of
the AI planner relate to the components present in many other
hierarchical planning approaches would be beneficial.

What elements differentiate an AI planner from other hierarchical
approaches that, at a high-level, also plan in terms of discrete,
abstract skills? Intuitively, and in layman's terms, what do each of
the referenced components (operators, ground atoms, lifted atoms, add
effects, delete effects, etc.) of the AI planner do and how do they
fit into the system as a whole? Elaborating on these points prior to
and/or within section 2 would make it much easier to follow what all
the symbols mean and remember their significance throughout the rest
of the paper.

The term "operator" is not described in layman's terms.

Line 22: "We are motivated by cases where it is easier to design [] or learn [] symbols than it is
to design skills."  What are examples of these cases? It would be
helpful if the authors could elaborate on why it may be easier to
learn/design symbols than it is to design skills. For humans, many of
us can learn skills (riding a bicycle, throwing a pot on a potter's
wheel) for which we cannot easily articulate predicates (features we are
paying attention to) or how we do the task. Most bicycle riders are
unaware of the non-minimum phase nature of bicycle dynamics and
that you initially turn the front wheel in the opposite way you wish to
ultimately turn.

Line 112: What are the limits of add/delete effects? Do they limit
what can be done with an attribute and its value in any way?

Line 172 - 182: The concept of "lifting" is described in detail for
the first time after already being referenced multiple times.

Line 322: "To the best of our knowledge, our work is the first to
learn operators, samplers, and policies that are compatible with
search-then-sample TAMP [29, 30, 31]."  What makes the learned
operators, samplers, and policies compatible with search-then-sample
TAMP; and why has this not been the case in other works? To make this
claim, it should be clear what compatibility means and how it is
measured.

The result on the doors environment suggests that success it not possible without
the manually added motion planning skill. An ability to recognize where a planning
skill is needed, and even an ability to default to a explicit planner in this case would
strengthen the result.



**Summary Of Recommendation:**

Ignoring philosophical differences with	the approach, I	think my
concerns focus on whether the evaluation is doomed to succeed.
The predicates are carefully chosen. Additional	symbolic
information is provided	by defining object (and value) types.
Not a lot of irrelevant	information has	been provided.
The simulations	have been greatly simplified. In this
kind of	work the scale or numbers of X (X = predicates,	objects, tasks,
demonstrations,	...) has to be large for the work to be
convincing.
The planning algorithm presented is clearly inefficient	and will
not scale well.

POST REBUTTAL COMMENT

I am still in favor of accepting this paper.

---

> ### Author Response · Authors · 2022-08-18
> **Thanks for your review!**
>
> Thank you for an incredibly thorough and thought-provoking review!
>
> >I disagree with the philosophy that abstractions should be correct representations of the underlying problem and should be taken seriously. I prefer approaches where abstract plans are often not executable but suggest where to search in lower level planning spaces… The role of abstraction may be to merely suggest good starting points or search directions for low level search.
>
> We actually very much agree with this perspective. Our view, too, is that the role of abstract plans should be to _guide_ planning at the low-level, rather than to supplant it. KD1 and KD2 are in this spirit. KD1 says that abstract plans should be used to suggest a sequence of abstract states, which low-level planning should attempt to follow. KD2 says that following any particular abstract plan may fail, so we should have the ability to consider alternatives. Though we frame the paper in terms of skill learning, an alternative framing would be to say that we are learning search guidance in the form of abstract operators, samplers, and policies, which underscores the point that the abstractions need not be “correct representations'' to be useful in decision-making.
>
> >Another issue with this paper is the manual feature selection… Our experience with good old fashioned AI (GOFAI) suggests that it has been difficult to find the implementable predicates (symbols) for real world robotics in the past. Has anything changed?
>
> We agree that manually-designed features and predicates can only get us so far. One of the main changes with respect to early work is that there has since been substantial progress in machine learning, and specifically, in unsupervised learning of disentangled state representations for decision-making. For example, recent work has shown promise in learning object-centric state representations from lower level sensory inputs (images) [1, 2]. Other work has also demonstrated success in predicate learning [3, 4, 5]. The prospect of starting with a small number of general, manually-programmed features and predicates and learning to correct or supplement them seems increasingly feasible. In our long-term vision, we also anticipate learning residual models [6, 7] to compensate for cases where our structural biases are overly rigid.
>
> >Could the paper discuss when useful abstractions are likely to exist and be findable, before a problem is solved?
>
> Let us distinguish between two different kinds of useful abstractions: manually-designed predicates, and learned predicates. Then we can interpret your question in two different ways: (1) When should we invest effort in designing predicates? (2) When learning predicates and skills, should we learn predicates or skills first? In the final version of our system, we envision manually designing an initial set of predicates that are generic and straightforward to program. For example, contact-related predicates, and predicates representing common spatial relations between objects, would be both applicable in a wide range of domains (generic) and easy to program (assuming a good perception system). After this initial design step, we then envision learning to improve, remove, or supplement the initial predicate set using data. Our initial experiments in Appendix C.4 suggest the viability of learning predicates first, before skills. Finally, as hinted in L23-34 of the introduction, we envision the agent continually improving both its predicates and skills as it gains experience online. To that end, we need not assume that all useful abstractions can be learned before the agent begins to act; the abstractions should become increasingly useful throughout the agent’s lifetime.
>
> >A related question is whether a search in abstract space is guaranteed to find a successful low level policy that does exist… Is it true that a successful abstract plan (as in line 132) always exists when a successful low level policy exists?
>
> With certain assumptions on neural network learning and planning, the agent should be able to solve the tasks for which it already has a demonstration. For neural network learning, we would assume zero training error. For planning, we would assume that abstract planning is able to revisit every abstract plan infinitely many times (which would require a minor modification to our planner, as hinted in Footnote 2 of page 4). The remaining question is whether the learned operators can reproduce the abstract state sequences corresponding to the predicates applied to the demonstrations. Previous work [5] has shown that the operator learning method we use in this work does have that guarantee.

---

> > ### Author Response · Authors · 2022-08-18
> > **Thanks for your review! (continued)**
> >
> > Beyond the demonstrated tasks, whether or not a low-level plan can be found will depend on the predicates and training data. One failure mode is overly restrictive operator preconditions, potentially preventing the generation of an important abstract plan. Another issue is that the neural network samplers or policies could fail on inputs that are out of distribution. Our intuition is that these failure modes would become vanishingly unlikely in the limit of infinite training data.
> >
> > >A simple change to the work is to make all predicates from all tasks available to each task. Lots of irrelevant predicates should cause segmentation and partitioning to fail.
> >
> > First note that due to object typing, creating a shared universe of predicates would not impact performance, since the object types are currently disjoint between our environments. However, we agree with your concerns about irrelevant predicates. We conducted additional experiments to investigate their impact; please see the additional experiments PDF in the meta-reviewer response.
> >
> > >The same argument goes for object types and objects.
> >
> > We conducted two additional experiments to study the influence of irrelevant objects on system performance; please see the additional experiments PDF in the meta-reviewer response.
> >
> > >I note that demonstrations are generated with handcrafted environment-specific policies. If doing the demonstrations is a solved problem, why are we bothering to learn to do these tasks?
> >
> > We use environments where it is possible to handcraft expert policies to facilitate large-scale experimentation. Towards supporting the claim that our approach can be scaled to environments where demonstrations are collected by humans, we conducted an additional experiment with human demonstrations; please see the additional experiments PDF in the meta-reviewer response.
> >
> > >Much more information needs to be provided about when the system gives up on trying to attain a particular subgoal… The planning algorithm proposed in section 4 is likely to be very inefficient.
> >
> > The bilevel planner that we use in this work is a very simple version of search-then-sample task and motion planning [8]. There are more sophisticated planning techniques that use similar representations; for one example, see [9]. For each abstract plan that is generated by the abstract search (Algorithm 1, Line 2), for each step in the abstract plan, we attempt up to $N_{samples}$ times to sample parameters for the skill that achieve the next expected abstract state. After that number of attempts, we backtrack and sample again, until we have exceeded the sampling budget at all steps. We describe this in Section 3, but we agree that it is brief. We will elaborate, but also refer to previous works [10, 11] that use the same planner and where the descriptions are extended.
> >
> > >I agree that segmenting trajectories at contact state changes makes sense. However, it probably makes sense to have a hierarchy of segments.
> >
> > We agree! Hierarchical segmentation is something we would like to explore in future work.
> >
> > >The learned subgoal sampler as designed is unlikely to explore out-of-distribution parameters.
> >
> > We agree. Depending on the circumstances, this could be advantageous, or not; out-of-distribution data is generally problematic for imitation learning, but we do want to allow for some generalization. We also want to clarify that the screw example given in your extended comment sounds more like a case where the policy would need to generalize, rather than the sampler. The sampler is only responsible for proposing the subgoal state that the skill should reach, not how the policy should reach it.
> >
> > >How is the "true" goal of a demonstration known?
> >
> > In our problem setting, the goal is given as part of the task, but we understand your question to be about real-world deployment. Since goals are predicate-based, it is reasonable to suppose that demonstrations are accompanied by goal annotations. Such annotations are easier to collect, for example, than an image representing a goal state, which would require solving the task first. If goals are not provided, then we could make use of literature on goal inference (e.g., [12, 13]). This direction is consistent with our long-term agenda, but outside the scope of the present study.

---

> > > ### Author Response · Authors · 2022-08-18
> > > **Thanks for your review! (continued)**
> > >
> > > >Add and delete operators were designed for features that had true or false values, and does not seem like the most useful semantics for robots…
> > >
> > > There is strong precedent for using STRIPS operators with add and delete effects in robotics domains, dating back to early work with Shakey [14] and continuing into present-day task and motion planning [8]. One key benefit of such operators is that we can use efficient off-the-shelf AI planners [15]. However, we agree that the representation can be limiting, and various extensions of PDDL have attempted to compensate for these limitations. Extending our operator representation, learning, and planning algorithms towards more general forms (e.g., numeric-valued fluents) is an interesting direction for future work.
> > >
> > > >It is much easier to read this paper AFTER reading the supplementary material… Sections 3, 4, and 5 would be easier to read if a simple example was used throughout… Words like operator, ground atom, and lifted atom seem to be necessary to understanding sections 3-5, but aren't explained until section 5… The concept of "lifting" is described in detail for the first time after already being referenced multiple times…. Overloading lambda for object type and real valued features is confusing.
> > >
> > > Thank you! We appreciate this feedback. Please see our general response to the meta-reviewer for steps that we will take to improve readability.
> > >
> > > >It might be useful to note that KD1 implies some lookahead so later (soft and hard) constraints are taken into account. This is what trajectory optimization and backpropagation in time do.
> > >
> > > Thanks, this is a good suggestion. We will add the following sentence on L51: “This desideratum implies that lookahead may be necessary, where the choice of skill early in the plan should be influenced by soft or hard constraints later in the plan.”
> > >
> > > >I can't figure out what the tasks are just from Figure 1.
> > >
> > > Space permitting, we will add 1-2 sentence descriptions of each task in the Figure 1 caption. Otherwise, we will clarify the section in the paper where readers can refer for task descriptions.
> > >
> > > >In top-k planning, how are the candidate plans made non-trivially different? Such a planner could generate plans that have trivial differences.
> > >
> > > Good question -- “diverse planning” is an active area of research in the AI planning community, and is closely related to top-k planning. See [16, 17] for in-depth discussions.
> > >
> > > >Can a segment be mapped to more than one skill dataset?
> > >
> > > No, we follow the usual definition of “partition”, where each element belongs to exactly one set in the partition.
> > >
> > > >Humans learn with 1 or a few demonstrations. This approach seems to need hundreds of demonstrations. Why?
> > >
> > > Humans learn low-level motor skills from more than just 1 or a few demonstrations -- we may receive a small number of demonstrations, but then perform a substantial amount of trial-and-error learning to actually acquire the skills. It is therefore not surprising that we would need a larger amount of data to learn comparable low-level skills for robotics. We are interested in exploring a more human-like learning paradigm that mixes a small number of demonstrations with online learning (and are actively working on this). However, we do want to have the ability to learn skills offline from a moderate number of demonstrations, since such demonstrations can be feasibly collected in robotics settings, e.g., with parallelization, and with potentially fewer safety concerns than in online learning.
> > >
> > > >How is this different from various other hierarchical approaches in machine learning that simultaneously learn parameterized skills and/or how to sequence them?
> > >
> > > We address this briefly in the first paragraph of related work (L305-313) but will elaborate, space permitting. One way to understand the relationship between our approach and prototypical hierarchical learning approaches (e.g., [18, 19]) is that we are model-based at both the high and low levels, where other approaches are model-free at both levels. Our learned skills induce an abstract transition model, and we plan in that abstraction (the high level). Conditioned on an abstract plan, we plan through the environment transition model (the low level). Other approaches learn a subgoal-generating reactive policy (a high-level “manager” policy) and a subgoal-consuming reactive policy (a low-level “worker” policy). Additional related work that we discuss in Section 7 is in between these two, e.g., planning at the high level, but then executing reactively at the low level.
> > >
> > > >What elements differentiate an AI planner from other hierarchical approaches that, at a high-level, also plan in terms of discrete, abstract skills?
> > >
> > > This is a large question! The survey paper [8] provides some answers.

---

> > > > ### Author Response · Authors · 2022-08-18
> > > > **Thanks for your review! (continued)**
> > > >
> > > > >What makes the learned operators, samplers, and policies compatible with search-then-sample TAMP; and why has this not been the case in other works?
> > > >
> > > > We agree that “compatibility” here is vague. We will replace that sentence with the following: “To the best of our knowledge, our work is the first to learn operators, samplers, and policies for search-then-sample TAMP."
> > > >
> > > > >An ability to recognize where a planning skill is needed, and even an ability to default to an explicit planner in this case would strengthen the result.
> > > >
> > > > One case that we discuss in the limitations section (L227-229) is where objects in the skill scope are insufficient to learn performant samplers and policies. Motion planning is an extreme example, because it requires collision-checking with all objects in the worst case. As mentioned in the text, there are avenues that we could pursue to learn samplers and policies that are functions of all objects in the scene, potentially replacing motion planning with a learned skill. However, in general, we would advocate for using generic and well-understood skills (like those based on motion planning) when they are available, and learning domain-specific skills otherwise. We would also like to have the ability to automatically determine when skill learning is likely to succeed, based on features of the task; this is an interesting direction for future work.
> > > >
> > > > >See what happens if you don't filter out rarely used operators
> > > >
> > > > We ran this additional experiment; Please see the additional experiments PDF in the meta-reviewer response.
> > > >
> > > > >See what happens if you don't reject out of bounds subgoals
> > > >
> > > > The “Samples=1” ablation in the main results provides insight on this question. Figure 4 shows that this ablation performs substantially worse than our main approach. If Samples=1 were to succeed, then we would not need to perform the abstract state check and backtracking search that we currently do. We expect that not rejecting out-of-bounds subgoals would thus perform similarly poorly.
> > > >
> > > > >Wider variety of simulated tasks - ideally non-symbolic tasks and not effectively pick and place: paint something? repair something with glue, tape, or string? shape something such as clay or dough?
> > > >
> > > > We agree that such environments would do well to validate and showcase our approach, and we will pursue this direction in future work. However, implementing these environments will require a very substantial engineering effort. We believe the four environments in the present paper are sufficient for a first demonstration of our approach. See also our response to Reviewer MQ6V for further discussion of the existing environments.
> > > >
> > > > >Explore several abstract plans in parallel rather than serially.
> > > >
> > > > This is a good idea and would certainly improve the efficiency of planning in many cases. We will leave this for future implementations, since our objective in this work is not to improve on the planner itself, but rather, to learn the representations necessary to use the planner.
> > > >
> > > > >Start with demonstrations alone, measured with low level sensors (not manually defined high level perceptions or predicates) as suggested in C.4.
> > > >
> > > > We want to clarify that Appendix C.4 does have some initial results that start with demonstrations alone (no predicates) and then uses the method of [5] to learn predicates before learning skills. We do not yet have results where the object features are also learned, e.g., from images [1, 2], but this is a priority for our future research.
> > > >
> > > > >In partitioning grouping segments if their effects are the same “up to object substitution” is unclear. Does this mean any object of any type can be substituted into a predicate or does it mean only different object instances within the same type can be substituted.
> > > >
> > > > Only object instances with the same type can be substituted. So, for example, we would never consider grounding a predicate like On(?x - block, ?y - block) with arguments that have type “robot”. We will clarify this in L166-167.

---

> > > > > ### Author Response · Authors · 2022-08-18
> > > > > **References**
> > > > >
> > > > > [1] Xu, D., Mandlekar, A., Martín-Martín, R., Zhu, Y., Savarese, S., & Fei-Fei, L. (2021) "Deep affordance foresight: Planning through what can be done in the future." ICRA.
> > > > >
> > > > > [2] Wang, C., Xu, D., & Fei-Fei, L. (2022). Generalizable Task Planning through Representation Pretraining. arXiv preprint arXiv:2205.07993.
> > > > >
> > > > > [3] Asai and C. Muise. Learning neural-symbolic descriptive planning models via cube-space
> > > > > priors: the voyage home (to strips) (2021). IJCAI.
> > > > >
> > > > > [4] Ahmetoglu, M. Y. Seker, J. Piater, E. Oztop, and E. Ugur (2020). Deepsym: Deep symbol generation and rule learning from unsupervised continuous robot interaction for planning. arXiv preprint arXiv:2012.02532.
> > > > >
> > > > > [5] Silver, R. Chitnis, N. Kumar, W. McClinton, T. Lozano-Perez, L. P. Kaelbling, and J. Tenenbaum (2022). Inventing relational state and action abstractions for effective and efficient bilevel planning. RLDM.
> > > > >
> > > > > [6] Silver, T., Allen, K., Tenenbaum, J., & Kaelbling, L. (2018). Residual policy learning. arXiv preprint arXiv:1812.06298.
> > > > >
> > > > > [7] Ajay, A., Wu, J., Fazeli, N., Bauza, M., Kaelbling, L. P., Tenenbaum, J. B., & Rodriguez, A. (2018). Augmenting physical simulators with stochastic neural networks: Case study of planar pushing and bouncing. IROS.
> > > > >
> > > > > [8] Garrett, C. R., Chitnis, R., Holladay, R., Kim, B., Silver, T., Kaelbling, L. P., & Lozano-Pérez, T. (2021). Integrated task and motion planning. Annual review of control, robotics, and autonomous systems, 4, 265-293.
> > > > >
> > > > > [9] Srivastava, S., Fang, E., Riano, L., Chitnis, R., Russell, S., & Abbeel, P. (2014). Combined task and motion planning through an extensible planner-independent interface layer. ICRA.
> > > > >
> > > > > [10] Silver, T., Chitnis, R., Tenenbaum, J., Kaelbling, L. P., & Lozano-Pérez, T. (2021). Learning symbolic operators for task and motion planning. IROS.
> > > > >
> > > > > [11] Chitnis, R., Silver, T., Tenenbaum, J. B., Lozano-Perez, T., & Kaelbling, L. P. (2022). Learning neuro-symbolic relational transition models for bilevel planning. IROS.
> > > > >
> > > > > [12] Ramírez, M. and Geffner, H. (2009). “Plan recognition as planning.” IJCAI.
> > > > >
> > > > > [13] Zhi-Xuan, T., Mann, J., Silver, T., Tenenbaum, J., & Mansinghka, V. (2020). Online Bayesian goal inference for boundedly rational planning agents. NeurIPS.
> > > > >
> > > > > [14] Nilsson, N. (1984). Shakey the robot. Technical Report.
> > > > >
> > > > > [15] Helmert, M. (2006). The fast downward planning system. JAIR.
> > > > >
> > > > > [16] Katz, M., & Sohrabi, S. (2020). Reshaping diverse planning. AAAI.
> > > > >
> > > > > [17] Katz, M., Sohrabi, S., & Udrea, O. (2020). Top-quality planning: Finding practically useful sets of best plans. AAAI.
> > > > >
> > > > > [18] Dayan, P., & Hinton, G. E. (1992). Feudal reinforcement learning. NeurIPS.
> > > > >
> > > > > [19] Nachum, O., Gu, S. S., Lee, H., & Levine, S. (2018). Data-efficient hierarchical reinforcement learning. NeurIPS.

---

> > ### Comment · Reviewer_Y3Ww · 2022-08-26
> > **What is the point of rebuttals?**
> >
> > There are 3 products of this review process.
> > 1) an improved paper
> > 2) possibly an improved supplemental-materials/appendix
> > 3) the reviews on OpenReview.net.
> >
> > I would argue that although 2 and 3 are publicly available, most future readers will not even be aware of their existence. From my point of view, only the paper (1) matters.
> >
> > Based on this reasoning, only changes to the paper matter. Responses to the reviewer that are not reflected by changes to the paper are a waste of time. In my view, every reviewer comment should either lead to a change in the paper, or simply be dismissed or ignored by the authors. This "rebuttal" has too much material that is not reflected by changes to the paper. I urge the authors to take the point of view that reviewers are typical readers, and other readers will have the same thoughts and questions as they read the paper. Change the paper, don't just have a conversation with the reviewers.

---

> > > ### Author Response · Authors · 2022-08-26
> > > **Re: What is the point of rebuttals?**
> > >
> > > Thanks for your reply. To clarify, we cannot change the paper during this rebuttal period. E.g., from an FAQ email from the chairs: "Q: Can I edit my original submission PDF? A: Not during the rebuttal phase." However, we did outline very specific plans to improve the paper based on the feedback from you and the other reviewers: in addition to the sentence-level changes that we listed in the "Readability Improvements" response above, our individual responses to you and the other reviewers include several other concrete changes that we will make to the main paper. Finally, we will include in the main paper a reference and brief summary of the additional experiments (with details in the appendix) that we presented in the reply to the meta-reviewer.

---

> > > > ### Comment · Reviewer_Y3Ww · 2022-08-26
> > > > **Can submit revised paper as additional file**
> > > >
> > > > Other authors have submitted revised papers as zip files. I don't know how they did this. In the past reviewers saw revised papers as part of this decision process. I agree this year's process with submitted revised papers being public is irrational, but since your paper is on arxiv already, I don't think you care about secrecy.

---

> > > > > ### Author Response · Authors · 2022-08-26
> > > > > **Re: Can submit revised paper as additional file**
> > > > >
> > > > > We interpreted the message from the chairs quoted above to mean that we should not edit the paper itself. Since the rebuttal phase is ending very soon, we will probably not have time to clarify the rules and post another draft of the paper (if it is allowed). We did our best to convey a clear vision for the next version of the paper with our original responses. If there are any points that we can make more concrete while the response window is still open, please let us know.

---

> > > > > > ### Comment · Reviewer_Y3Ww · 2022-08-26
> > > > > > **policy**
> > > > > >
> > > > > > I agree the communications have been confusing. I hear you that you are out of time.
> > > > > >
> > > > > >
> > > > > > The PC wanted to preserve the original submission, so comments about it remain valid. This was not intended to preclude submitting a revised version. You should have gotten this email:
> > > > > >
> > > > > >
> > > > > > CoRL2022: Rebuttal instructions
> > > > > >
> > > > > > OpenReview <noreply@openreview.net>
> > > > > >
> > > > > > Fri, Aug 19, 2:30 PM (7 days ago)
> > > > > >
> > > > > > Dear Author,
> > > > > >
> > > > > > We would like to provide the following instructions for the CoRL rebuttal/discussion phase since the process might be unfamiliar to some of you.
> > > > > >
> > > > > > Responding to reviewer comments: You can respond to each reviewer individually using the “Comment with Files”. You will be able to comment and optionally choose to upload a file in the zip or pdf format. You can use this feature to upload the revised paper, though you are not required to if the new changes can be communicated clearly with the reviewers via comments or an upload of a new image, table, result, etc. Note that all files and comments will be open to the public.
> > > > > >
> > > > > > Limitations on uploads: If you choose to upload the revised paper during the rebuttal phase, you do not need to worry about going over the 8-page limit.
> > > > > >
> > > > > > Anonymity requirements: Important! It is critical to continue to maintain anonymity. Please be careful with communications and uploads. Anonymity violations can result in desk rejection.
> > > > > >
> > > > > > Etiquette: Please also be mindful that reviewers have a lot of work on their plate. Keeping your comments and uploads concise and to the point will be appreciated.
> > > > > >
> > > > > > Private Communications: If you wish to make a private comment to ACs or reviewers, you could use the “Official Comment” button, which allows you to select the intended readers.
> > > > > >
> > > > > > The PC team

---

### Official Review · Reviewer_ZRBu · 2022-08-05

**Originality:** Good
**Technical Quality:** Very Good
**Clarity Of Presentation:** Very Good
**Impact:** 3

**Recommendation:**

Weak Reject: I recommend rejecting the paper, but will not argue for my recommendation if the majority of other reviewers have a different opinion.

**Summary:**

This paper exploits pre-defined symbols to discover skills that enable symbolic and low-level plan generation. In an environment with continuous states and actions, assuming pre-defined symbols, symbol mappings, predicates, object ids, transition functions, and unsegmented demonstrations, the proposed system discovers skills that encode symbolic operators (precondition and effect predicates) and skill policies.  Importantly, the skills that encode <precondition, effect> symbols can be achieved via several low-level goals, which brings flexibility in planning: The system can produce multiple symbolic plans in the outer loop and checks out with which detailed policy the symbolic plan is possible in the inner loop. The method is sound, verified in a number of tasks, and compared against baselines. The problem is an important problem, assuming pre-defined symbols and building up everything on top of the pre-defined symbols bring limitations to the scalability of the method.

**Issues:**

- Please clarify the detailed contribution with respect to [11] and [12] in algorithmic and implementation levels.
-  The writing of the paper is clear. However, I am not sure if it requires such extensive and a bit complicated notation. Is it possible to simplify to improve the accessibility to a wider audience?
- The description of KD2 can be improved. It is not clear whether creating multiple symbolic plans or the flexibility in the inner loop is more important.
- Please clarify whether object types were predefined or automatically generated from the beginning and also in 5.1 Lifting subsection.
- In Policy Learning, please provide more details (possibly in the Appendix). Is only u_i learned/predicted? How about u_{i+1:k}, i.e. the parameters of the action execution? Why/how a parameter from a single (arbitrary) time-step represents the policy? Please clarify.
- Is it possible to learn a skill with pre-conditions and effects that include a variable number of objects? As far as I see it is not possible. Is it an important limitation? How to extend the method to address this?
- In the experiments, what happens if the training tasks include 3-4 buttons/cups and test tasks include 2-3 buttons? As far as I see, all 3 object types are required to be included in the preconditions (as they are all common in all training data), and therefore the method will fail with 1-2 objects. Please clarify whether this is the case, discuss whether this is an assumption, and re-discuss the generalization capability of the method taking into account this.
- Achterhold et al. (2022) also learn symbolic operators along and continuous control, but with an RL agent. They also do hierarchical planning. This work can be included in related work or discussion

Achterhold, Jan, Markus Krimmel, and Joerg Stueckler. "Learning Temporally Extended Skills in Continuous Domains as Symbolic Actions for Planning." arXiv preprint arXiv:2207.05018 (2022).


**Quality Of The Limitations Section:**

Limitations are addressed clearly

**Reviewer Expertise:**

4: The reviewer is confident but not absolutely certain that the evaluation is correct

**Robotics Focus:**

Highly relevant to robotics but no hardware experiments

**Strengths And Weaknesses:**

It is very challenging, important (and timely) to learn skills that allow robots to make both high-level (symbolic) and low-level reasoning from the continuous sensorimotor experience of the robot. High-level reasoning requires symbols and operators to be learned, and low-level reasoning and execution require learning of detailed flexible policies that are linked with the operator definitions. The authors well-address the latter topic, and in this sense, their contribution is important and significant. However, they need to clarify their contribution with respect to [11] and [12] as detailed below.

Saying this, the differences between the current work and the previous work [11] and [12] are not properly provided. The main contribution is stated as learning policies compared to [11] & [12] where the parametrized policies were pre-defined in the previous work. Stating this in only one line and then describing the Neuro-Symbolic Skills (Section 3) and Bilevel Planning (Section 4) without much referring to it is misleading about the contribution. In [12], Chitnis et al. also partitions skill data, unifies transitions based on symbolic effects, find the preconditions and effects of operators, and learn action sampler. This roughly corresponds to 5.1 and operator learning and sampler learning parts of 5.2 in the current paper. I might be missing the details, however, the only difference of the current paper seems to be in step 5.2 - policy learning; whereas all other sub-parts were realized in some form in [11] and [12].

Therefore, the authors need to explicitly and carefully emphasize the exact contribution at the algorithmic level and at the realization/implementation level. They also need to provide why [11] or [12] were not included in the experiments as baselines. Maybe they might not be directly applicable for different reasons, then these reasons should be clearly pointed out. Additionally, was it not possible to apply [11] and [12] in this problem setting with small modifications? This also needs to be explained.




**Summary Of Recommendation:**

This paper proposed, implemented and verified a sound method for a challenging and important problem. However, it looks incremental in its current form.

---

> ### Author Response · Authors · 2022-08-18
> **Thanks for your review!**
>
> >Please clarify the detailed contribution with respect to [11] and [12] in algorithmic and implementation levels.
>
> In brief, [12] extends [11] by removing the assumption that samplers are given. Our work here extends [12] by removing the assumptions that high-level controllers are given (e.g., pick, place, move), and that demonstration data is provided in terms of those high-level controllers. We mention this relationship on L66-67, L155-156, and L317-318, but we can expand these sections to make the contribution more clear.
>
> Removing the assumption that high-level controllers are given requires several nontrivial steps. First, in [11, 12], the demonstration data is given in terms of high-level controllers: each transition corresponds to the execution of an entire controller, and the controller identity is known. This setting makes operator learning much easier because each transition corresponds to exactly one operator. The controller identity is also used to make operator learning easier. In contrast, we are given demonstration data where the actions are low-level (e.g., end effector positions of the robot), and we seek to learn operators that correspond to the execution of _many_ low-level actions in sequence. This necessitates segmentation (Section 5.1). Second, because the controllers are fully defined in the previous work, including their continuous parameterizations, it is straightforward to set up the sampler learning problems. In contrast, we have no such continuous controller parameterization given to us. One of our main insights is that subgoal states can be used as the basis for continuous parameterization. This insight follows from KD1 and has the benefit that we can automatically derive targets for learning from the demonstration data (Section 5.2, Sampler Learning). Finally, we must learn the controllers themselves (Section 5.2, Policy Learning). In the previous work, these controllers were hardcoded.
>
> In response to the request for an empirical comparison to the previous work, we also conducted an additional experiment; please see the additional experiments PDF in the meta-reviewer response.
>
> >The writing of the paper is clear. However, I am not sure if it requires such extensive and a bit complicated notation. Is it possible to simplify to improve the accessibility to a wider audience?
>
> Thanks for the feedback -- please see our general response to the meta-reviewer post with our  concrete plan to reduce notation and improve accessibility.
>
> >It is not clear whether creating multiple symbolic plans or the flexibility in the inner loop is more important.
>
> Questions Q4 and Q5 are posed at the beginning of Section 6 to highlight how our experiments evaluate the importance of KD1 and KD2, respectively. The importance of KD1 is reflected most clearly in the Samples=1 ablation, and also in the No Subgoal baseline. The importance of KD2 is reflected most clearly in the Abstract Plans=1 ablation, and also in the GNN Meta baseline. The results in Figure 4 show that ablating KD1 (orange and red) leads to worse performance than ablating KD2 (blue and gray) in Cover, while the opposite holds true in Stick Button and Coffee. Thus, which of KD1 or KD2 ends up being more important depends on the task and environment. KD1 seems to be more important in Cover because most abstract plans are reasonable (it does not matter in which order you place blocks on their respective targets), while where you grasp a block determines if covering its target will be possible. KD2 seems to be more important in Stick Button because the order in which you press buttons or pick up the stick matters a lot (some buttons are not reachable without the stick), while most buttons further away are still reachable with the stick even if you don’t grasp the stick such that you have maximal reach with it. In other words, in some situations the order of abstract states visited matters more than the specific low-level states you reach in them, while in other situations, the opposite is true.
>
> >Please clarify whether object types were predefined or automatically generated from the beginning and also in 5.1 Lifting subsection.
>
> The object types are predefined. We will articulate this more clearly in Section 2 and emphasize it in Section 5.1 where it’s relevant to how lifting is performed.

---

> > ### Author Response · Authors · 2022-08-18
> > **Thanks for your review! (continued)**
> >
> > >In Policy Learning, please provide more details (possibly in the Appendix). Is only u_i learned/predicted? How about u_{i+1:k}, i.e. the parameters of the action execution? Why/how a parameter from a single (arbitrary) time-step represents the policy? Please clarify.
> >
> > Recall from Section 5.1 that a segment is composed of a sequence of low-level states and the low-level actions taken between each of the low-level states in that sequence, and that we partition the set of all segments into skill datasets. All the segments in a given skill dataset have the same abstract effects (equivalent up to object substitution) and an operator is induced for each skill dataset. We learn a policy for each operator that takes as input (1) the low-level state of each of the objects in the skill scope and (2) a subgoal (a proposed low-level state of each of the objects in the skill scope that should hold when the policy terminates) and outputs a low-level action. We use all the segments in a skill dataset to construct a dataset of input-output pairs for supervised learning of a subgoal-conditioned policy. At test time, at each time step, the input is constructed from the current state and the sampler’s suggested subgoal (which is suggested only just once at the beginning of the policy’s execution) and passed into the neural network that represents the policy. The network predicts a low-level action to take at that time step. A “parameter from a single (arbitrary) time-step” does not represent the policy.
> >
> > For example, a particular segment contains some states (x_{j-1}, …,  x_{k}) and actions (u_j, …, u_{k-1}). To create our input-output training data for the policy from this segment, we construct
> > (x’_{i-1} concatenated with x’_{k-1}, u_i) for each i in j<=i<=k, where we let x’ indicate the vector obtained by concatenating the low-level state vector for all the objects in the skill scope. Each u_i for j<=i<=k is an output target for the neural network, so u_{i+1:k} are not parameters of any kind; they are merely targets for the regression problem.
> >
> > >Is it possible to learn a skill with pre-conditions and effects that include a variable number of objects? As far as I see it is not possible. Is it an important limitation? How to extend the method to address this?
> >
> > Our current approach does not learn skills that handle a variable number of objects. Our method would create a separate skill for each different number of objects in the preconditions and effects. For example, if we were trying to learn a skill to pick up a clump of small pieces of candy, we would create a skill for picking up a clump of three pieces of candy separately from a skill for picking up a clump of four pieces of candy. This is not ideal, as the policy for these actions is basically the same. This problem is complicated by the fact that you might actually want different skills when handling different numbers of the same object. For example, a policy for picking up a single piece of candy (two-finger or three-finger motion) vs. a clump of five pieces of candy (using entire hand) vs. a clump of twenty-five pieces of candy (using both hands) would likely be quite different, so you might want a separate skill for each of these. As another example, a policy for juggling three items might be pretty different from a policy for juggling ten items. Examples for a variable number of items in the preconditions tend to not be as drastic and tend not to involve a change in the policy itself.
> >
> > One approach to enable our skill to handle a variable number of objects would be to group segments that see the same precondition or effect applied to multiple objects of the same type together, and then to parameterize the policy in a way that can handle a variable number of objects, such as with a graph neural network.
> >
> > The importance of being able to handle a variable number of objects depends on the extent to which you can get by without handling that: what tasks necessitate handling a number of objects together? For example, if there were one hundred pieces of candy on the ground and your goal was to put each piece of candy in a bucket, picking up one piece of candy at a time might be okay. However, if your goal was to tie a piece of string around a clump of sticks, there would be no way to do this one-by-one, and you’d need a way to learn a policy that achieves this for a variable number of sticks.

---

> > > ### Author Response · Authors · 2022-08-18
> > > **Thanks for your review! (continued)**
> > >
> > > >In the experiments, what happens if the training tasks include 3-4 buttons/cups and test tasks include 2-3 buttons? As far as I see, all 3 object types are required to be included in the preconditions (as they are all common in all training data), and therefore the method will fail with 1-2 objects. Please clarify whether this is the case, discuss whether this is an assumption, and re-discuss the generalization capability of the method taking into account this.
> > >
> > > How the preconditions are determined for an operator is discussed under operator learning in Section 5.2. It is not true that “all 3 object types are required to be included in the preconditions” for an operator (but we may be misunderstanding your point). It is generally easier to go from training on more objects to testing on fewer objects, as compared to the converse. We provide an experiment to illustrate this; please see the additional experiments PDF in the meta-reviewer response.
> > >
> > > >Achterhold et al. (2022) also learn symbolic operators along and continuous control, but with an RL agent. They also do hierarchical planning. This work can be included in related work or discussion
> > >
> > > Thanks for the suggestion! This is an interesting paper that we will mention in the related work as an example of applying hierarchical planning to long-horizon problems and learning symbolic transition models and skills. To contrast their work with ours, they do not address KD1 or KD2, which are both critical in long-horizon robotics problems like the ones in our experiments.

---

### Meta-Review · Area_Chair_8Hf8 · 2022-08-12

**Recommendation:** Accept (Poster)
**Confidence:** 4

**Metareview:**

Combining learning of symbolic high-level and operational low-level knowledge is an important and useful research direction, and bringing their learning together is a logical and potentially impactful idea. The paper proposes a sound method for this challenging task. The paper is well written and presented, although some explanation could be potentially simplified and required information is only found in the appendix. The presented results are very promising, especially wrt. to the generalization in the ‘low training data’ setup, and the provided ablations provide useful insights.

However, there are several concerns raised that should be considered and clarified. Mainly regarding clarification of some explanations and results, as well as the limitations of the assumptions and the clean/perfect simulations . See the following proposed action items based on the provided reviews (more details in the corresponding reviews).

Proposal of action items:
- Clarifying contribution wrt. to [11,12]
- Clarifying generalization abilities wrt. to number of objects
- Discussing the importance of carefully hand-designed features/predicates
- Discussing what properties an abstraction must have to find successful solutions
- Discussing the sim-to-real gap, i.e., if the approach is (too) tailored to clean simulations
- Discussing the effect of KD2 on planning efficiency, and how to find really different abstract plans
- Consider moving important information from the appendix into the paper
- Consider used words and definition earlier, simplifying explanations
- Clarifying ‘compatibility’ wrt. TAMP
- Consider experiment(s) (or discussion) with more ‘noise’ (irrelevant predicates, objects, operators, noisy observations,..)

-----

I’d like to thank the reviewers and authors for the constructive discussions and valuable updates. The overall recommendations and discussions are in favor of accepting the paper, hence, my recommendation is to accept the paper. However, I agree with Reviewer Y3Ww that only changes in the paper really matter, and the authors are highly encouraged to include their answers and changes in the final updated manuscript.

**Best Paper Nomination:**

No

---

> ### Author Response · Authors · 2022-08-18
> **General Response**
>
> Thanks very much to all reviewers and the meta-reviewer.
>
> ### New Experiments
>
> In response to the reviewer feedback, we conducted 7 additional experiments. Please see the attached PDF.
>
> ### Action Items
>
> Thanks very much to the meta-reviewer for suggesting main action items. We will mention below where we have completed each of these items.
>
> >Clarifying contribution wrt. to [11,12]
>
> See our response to Reviewer ZRBu and additional experiments in the attached PDF.
>
> >Clarifying generalization abilities wrt. to number of objects
>
> See our response to Reviewer ZRBu and additional experiments in the attached PDF.
>
> >Discussing the importance of carefully hand-designed features/predicates
>
> See our response to Reviewer Y3Ww.
>
> >Discussing what properties an abstraction must have to find successful solutions
>
> See our response to Reviewer Y3Ww.
>
> >Discussing the sim-to-real gap, i.e., if the approach is (too) tailored to clean simulations
>
> See our response to Reviewer Y3Ww.
>
> >Discussing the effect of KD2 on planning efficiency, and how to find really different abstract plans
>
> See our response to Reviewer Y3Ww.
>
> >Consider moving important information from the appendix into the paper
>
> Space permitting, we will add 1-2 sentence descriptions of each task in the Figure 1 caption. Otherwise, we will clarify the section in the paper where readers can refer for task descriptions. The readability improvements suggested below describe how we will provide some more running examples throughout the text so that readers don’t need to look at the Appendix to see examples.
>
> >Consider used words and definition earlier, simplifying explanations
>
> See our plan to improve readability below.
>
> >Clarifying ‘compatibility’ wrt. TAMP
>
> See our response to Reviewer Y3Ww.
>
> >Consider experiment(s) (or discussion) with more ‘noise’ (irrelevant predicates, objects, operators, noisy observations,..)
>
> See the additional experiments in the attached PDF.

---

> > ### Author Response · Authors · 2022-08-18
> > **Readability Improvements**
> >
> > ### Readability Improvements
> >
> > Reviewers suggested that we reduce notation and add more examples. To this end, we now describe our concrete plan to improve readability, and also respond to specific comments:
> > * In general, we agree that too much notation can be overwhelming, but we also found when having colleagues proofread early drafts that a lack of notation led to confusion about things such as what exactly the domain and range of particular functions were, what was precisely was contained in components like a “skill”, an “operator”. We will try to strike a better balance and try to make every definition contain both sufficient English description and also some math, such as:
> >    * L83-84: “The transition function is a known, deterministic mapping from a state and action to a next state, denoted [math expression]."
> > * >Y3Ww: “Overloading lambda for object type and real valued features is confusing”
> >    * We are not overloading \lambda on L79-80. We use \lambda to indicate an object type and say that such an object has a tuple of dim(\lambda) real-valued features. To be more clear, we will say “... and a tuple of real-valued features of dimension dim(\lambda)” on L80.
> > * >Y3Ww: “Line 172 - 182: The concept of "lifting" is described in detail for the first time after already being referenced multiple times.”
> >    * In Section 2, “predicate”, “ground atom”, and “lifted atom” are currently defined. We do not discuss lifting before section 5; we only use the term “lifted atom” which has already been defined. We do mention “lifting” in the first paragraph of Section 5 before we define it, but that is in a topic sentence that serves to outline for the reader what we are about to define below in the “Lifting” subsection. However, we will add the following sentence in Section 2 to improve clarity: “When we perform _lifting_, we replace objects with variables that have the objects’ types”.
> > * We agree that a running example will improve clarity. We currently introduce Stick Button in Section 1 and continue it into Section 2. Space permitting, we will also extend the example into Sections 3, 4, and 5.
> > * Toward reducing and clarifying notation:
> >    * In L95,96 we will add more English description: “a solution to a task is a sequence of actions that achieves the goal and does not exceed the task horizon, i.e. [math expression].”
> >    * In L98,99 we will replace “$\mathcal{T}$” with “the task distribution”
> >    * In L101,102, we will replace the mathematical description for “$\overline{x}$” with “... where $\overline{x} = (x_{0}, …, x_{l})$ is the sequence of states visited”.
> >    * In L112,113 we will replace symbols $\Psi$ and $\overline{v}$ with English via “... each a set of lifted atoms over the predicates and arguments.”
> >    * In L113,114  we will change the definition to “A subgoal-conditioned policy takes as input a tuple of objects corresponding to the skill’s arguments, the current state $x$, and a subgoal state $x’$, and maps it to an action, denoted [math expression]”.
> >    * We will state explicitly that underlines are used to denote grounded quantities.
> >    * In L117, we will update the definition for a ground skill to be “A ground skill is a skill with objects that match its arguments, denoted [math expression].”
> >    * In L118, we will update the definitions as “The corresponding ground operator, ground policy, and ground sampler are defined similarly: with a tuple of objects that match the typed variables, denoted [math expression].
> >    * In L120 we will update the definition to be “A grounded skill induces a transition in the abstract state space via the effects of its ground operator. The abstract transition function maps an abstract state and a ground skill to abstract state, denoted [math expression].
> >    * We will add clarification to L124: “... the sampler should generate a subgoal [math expression] that satisfies the skill’s effect, i.e. [math expression], and the actions… “.
> >    * In L132, we will update the definition of an abstract plan to “An abstract plan … is a sequence of ground skills that achieves the goal, i.e. [math expression].
> >    * In L160 we will add: “induces a segment, … , a subsequence of consecutive actions, denoted [math expression], …”
> >    * In L164 we will get rid of the mathematical definition of “$\Gamma$” and describe the quantity in English.
> >    * In L168 we will clarify the symbol for set subtraction.
> >    * In L169-170, we will update the explanation to “We define two segments to be equivalent if there exists an injective mapping between their affected object sets and where each object in $O_{1}^{e}$ shares the type of the object it is mapped to in $O_{2}^{e}$. Equivalent segments are placed into the same skill dataset.”
> >    * In L192 we will clarify, “The preconditions P are then the intersection of lifted atom sets for each segment, i.e. [math expression], …”.